# The inherent flexibility of receptor binding domains in SARS-CoV-2 spike protein

**Hisham M Dokainish[1], Suyong Re[2,3], Takaharu Mori[1], Chigusa Kobayashi[4], Jaewoon Jung[1,4], Yuji Sugita[1,3,4]***

[1]Theoretical Molecular Science Laboratory, RIKEN Cluster for Pioneering Research, Wako, Japan; [2]Artificial Intelligence Center for Health and Biomedical Research, National Institutes of Biomedical Innovation, Health and Nutrition, Osaka, Japan; [3]Laboratory for Biomolecular Function Simulation, RIKEN Center for Biosystems Dynamics Research, Kobe, Japan; [4]Computational Biophysics Research Team, RIKEN Center for Computational Science, Kobe, Japan

**Abstract** Spike (S) protein is the primary antigenic target for neutralization and vaccine development for the severe acute respiratory syndrome coronavirus 2 (SARS-CoV-2). It decorates the virus surface and undergoes large motions of its receptor binding domains (RBDs) to enter the host cell. Here, we observe Down, one-Up, one-Open, and two-Up-like structures in enhanced molecular dynamics simulations, and characterize the transition pathways via inter-domain interactions. Transient salt-bridges between $RBD_A$ and $RBD_C$ and the interaction with glycan at $N343_B$ support $RBD_A$ motions from Down to one-Up. Reduced interactions between $RBD_A$ and $RBD_B$ in one-Up induce $RBD_B$ motions toward two-Up. The simulations overall agree with cryo-electron microscopy structure distributions and FRET experiments and provide hidden functional structures, namely, intermediates along Down-to-one-Up transition with druggable cryptic pockets as well as one-Open with a maximum exposed RBD. The inherent flexibility of S-protein thus provides essential information for antiviral drug rational design or vaccine development.

*For correspondence:
sugita@riken.jp

## Editor's evaluation

Using extensive molecular dynamics simulations with a novel enhanced sampling technique, the authors are able to characterize the structural flexibility of the SARS-CoV2 spike protein and identify new conformational states. These insights will be valuable to the design of novel strategies that modulate the interactions of the spike protein during the infection process.

## Introduction

The severe acute respiratory syndrome coronavirus 2 (SARS-CoV-2) has caused over 430 million infections and 5.9 million deaths, as of February 2022 (https://coronavirus.jhu.edu/map.html). It represents an urgent need for an effective medical intervention strategy to avoid further social and economic consequences (*Hu et al., 2021*). Different types of vaccines, for example, those from Pfizer-BioNTech or Moderna using the mRNA of spike (S) protein, are currently available, and there are several FDA approved drug candidates under consideration (*Li et al., 2021*; *Venkadapathi et al., 2021*; *Alvarado et al., 2022*). At the same time, more infectious mutants such as B.1.617.2 (Delta), B.1.427/B.1.429 (Epsilon), and more recently Omicron have appeared (*McCallum et al., 2021*; *Cao et al., 2022*), and some evade from the immune system (*Yurkovetskiy et al., 2020*; *Harvey et al., 2021*; *Wang et al.,*

*2021*; *Gobeil et al., 2021*). Furthermore, the virus's ability to infect a wide range of vertebrates, which could act as a reservoir, points out the future risk despite the vaccination progress (*Prince et al., 2021*). A deeper understanding of the virus molecular structure and infection mechanism is crucial to stop the virus transmission including mutant strains (*V'kovski et al., 2021*).

SARS-CoV-2, an enveloped positive single-stranded RNA virus, has a large genome of approximately 30 kb encoding 29 proteins (*V'kovski et al., 2021*; *Finkel et al., 2021*). A transmembrane homotrimeric class I fusion glycoprotein decorating the virus, known as S-protein, plays a critical role in the viral cell entry (*Ke et al., 2020*; *Shang et al., 2020*). In an immediate response to the pandemic, more than 300 S-protein structures determined with cryo-electron microscopy (EM) and X-ray crystallography have been reported and rapidly advanced in our understanding of the S-protein/receptor binding mechanisms (*Wrobel et al., 2020*; *Hsieh et al., 2020*; *Barnes et al., 2020a*; *Wrapp et al., 2020*; *Cai et al., 2020*; *Wang et al., 2020*; *Yan et al., 2020*). The N-terminal subunit (S1), which is composed of the N-terminal domain (NTD), the receptor binding domain (RBD), and two other subdomains (SD1 and SD2) (*Figure 1—figure supplement 1*; *Wrapp et al., 2020*; *Walls et al., 2020*), initially binds to the host cell receptor angiotensin-converting enzyme 2 (ACE2) (*Yan et al., 2020*). This binding is followed by priming of the C-terminal subunit (S2) and its large conformational changes leading to the membrane fusion for the cell entry (*Shang et al., 2020*; *Fan et al., 2020*). S-protein is covered by 66 N-glycans, 22 per protomer, to evade from the host cell immune system (*Grant et al., 2020*; *Watanabe et al., 2020*). To block the initial binding with ACE2, vaccines for stimulating the immune system or antibodies/small-molecule drugs for neutralizing the virus is the primary target for medical interventions. Numerous cryo-EM structures have revealed that RBDs can take Down or several Up forms including one-Up, two-Up, and three-Up conformations (*Figure 1a*; *Wrobel et al., 2020*; *Hsieh et al., 2020*; *Barnes et al., 2020a*; *Wrapp et al., 2020*; *Cai et al., 2020*; *Fan et al., 2020*; *Barnes et al., 2020b*; *Juraszek et al., 2021*). Up forms are accessible to the ACE2 receptor in the cell entry, while the Down is inaccessible (*Wrapp et al., 2020*; *Yan et al., 2020*; *Xu et al., 2021*). Note that most of the cryo-EM structures representing Up forms were determined together with other proteins, such as antibodies or a fragment of ACE2 (*Supplementary file 1A*, *Table 1*). A recent single molecule fluorescence resonance energy transfer (smFRET) experiment suggested that the RBD Down-to-Up transition occurs even without any ligand of S-protein, and at least one transient intermediate exists during the transition (*Lu et al., 2020*). These experiments suggest the potential flexibility of S-protein conformation and the possible involvement of multiple intermediates along transition pathways. The exploration of a wide conformational space of S-protein is necessary for characterizing the transition pathways, which give us better chances to find cryptic drug/antibody binding sites otherwise elusive.

Molecular dynamics (MD) simulations at the atomic level have been conducted to explore conformational dynamics of S-protein including the glycans at its surface, which are largely missed in the cryo-EM structures. Several microsecond-scale simulations showed the conformational flexibility of the stalk region in S-protein, different levels of glycan shielding between Down and Up, and their relevance in the ACE2 binding (*Grant et al., 2020*; *Turoňová et al., 2020*; *Choi et al., 2021*; *Pavlova et al., 2021*). They also identified specific interactions between side-chain residues or those with glycans to stabilize either Down or Up structure (*Casalino et al., 2020*; *Mori et al., 2021*). However, the timescale of the Down-to-Up transition in S-protein is far slower than that is attainable in the current MD simulations, leaving the inherent flexibility and transition pathways largely unknown. Therefore, several enhanced sampling methods including targeted MD (*Mori et al., 2021*), steering MD (*Gur et al., 2020*), nudged elastic band/umbrella sampling (*Simmerling et al., 2020*), two-dimensional umbrella sampling (*Pang et al., 2021*), adaptive sampling (*Zimmerman et al., 2021*), and weighted ensemble methods have been applied to investigate the RBD transition from Down to one-Up (*Sztain et al., 2021*). In these simulations, pre-defined reaction coordinates and/or bias potentials along the coordinates were used to enhance the motions of a single RBD, which limit the number of RBDs to be investigated simultaneously.

Here, we apply the generalized replica exchange with solute tempering of selected surface charged residues (gREST_SSCR) (*Dokainish and Sugita, 2021*), which can enhance domain motions of a protein by exchanging the solute temperature of selected surface residues between neighboring replicas like the original gREST method (*Kamiya and Sugita, 2018*). gREST_SSCR is distinct from the other simulations applied to S-protein in that no reaction coordinates as well as bias potentials are used, allowing us to examine the inherent flexibility of S-protein involving more than one RBD.

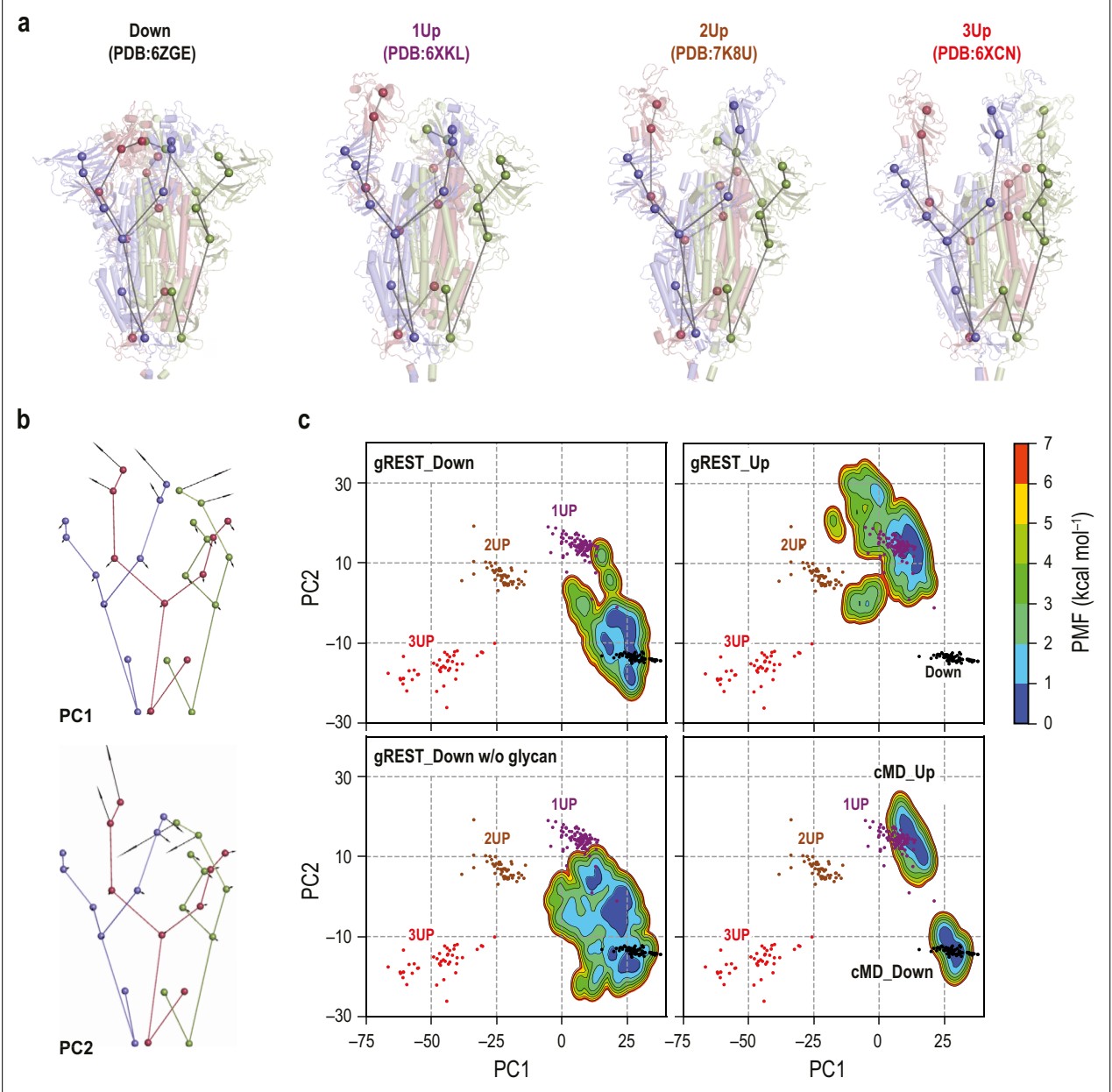

**Figure 1.** Comparisons of spike (S) protein structures in molecular dynamics (MD) simulations with cryo-electron microscopy (cryo-EM) structures. (**a**) Twenty-seven coarse-grained beads representations of four representative cryo-EM structures: Down (PDB ID: 6ZGE), 1Up (6XKL), 2Up (7K8U), and 3Up (6XCN). Chains A, B, and C in S-protein are shown in red, blue, and green, respectively. (**b**) The lowest two modes (PC1 and PC2) in principal component analysis of the 27-beads model of 289 cryo-EM structures. PC1 and PC2 respectively represent a symmetric and an anti-symmetric Down-to-Up motion of the receptor binding domains. The vector direction of PC1 is reversed so that it points from Down to Up for better visualization. The vectors are magnified 100 times for clarity. (**c**) Free-energy landscapes at 310 K along the PC1 and PC2 obtained from the simulations: (top) gREST_SSCR simulations with glycans starting from Down (500 ns) and Up (300 ns), (bottom left) gREST_SSCR simulation without glycans from Down (150 ns), (bottom right) conventional MD simulations with glycans starting from either Down or Up. The positions of cryo-EM structures are also shown for comparison. Wherein, Down, 1Up, 2Up, and 3Up conformations are shown in black, purple, brown, and red dots, respectively.

The online version of this article includes the following video and figure supplement(s) for figure 1:

**Figure supplement 1.** Structural model of the spike protein used in this study.

**Figure supplement 2.** Glycans in our spike protein models.

**Figure supplement 3.** Performance of the gREST_SSCR simulations.

**Figure supplement 4.** Characterization of the receptor binding domain (RBD) conformational change in the three gREST_SSCR simulations.

*Figure 1 continued on next page*

*Figure 1 continued*

**Figure supplement 5.** Analysis of the intra-domain stability of receptor binding domain (RBD) and N-terminal domain (NTD) in the gREST_SSCR simulations.

**Figure supplement 6.** Scheme of the protomer rotation and analysis of the rotated trajectories of the gREST_Down simulation.

**Figure supplement 7.** Free-energy landscape (FEL) along the hinge/twist angles in the gREST_Down w/o glycan and gREST_Up simulations.

**Figure 1—video 1.** Down-to-1Up transition of replica 16 in gREST_Down simulation.

https://elifesciences.org/articles/75720/figures#fig1video1

**Figure 1—video 2.** 1Up-to-2Up transition of replica 4 in gREST_Up simulation.

https://elifesciences.org/articles/75720/figures#fig1video2

**Figure 1—video 3.** Down-to-1Up transition of replica 6 in gREST_Down w/o glycan simulation.

https://elifesciences.org/articles/75720/figures#fig1video3

As listed in *Table 1*, the RBD motion in the RBD/SD1 monomer as well as the trimeric S-protein with and without glycans is systematically investigated. The intrinsic flexibility of S-protein observed in the enhanced sampling MD simulations suggests new mechanisms underlying the Down-to-Up transitions, glycan shielding for the binding with antibodies, and unprecedented cryptic pockets in the intermediates between Down and one-Up.

## Results

### Enhanced RBD motions are observed in gREST_SSCR simulations

*Table 1* lists all simulations performed in this study (for details, SI methods, *Figure 1—figure supplements 1 and 2*). Eight pairs of positively and negatively charged residues at RBD interface in each chain were selected as solute in gREST_SSCR and their Coulomb and Lennard-Jones interactions were scaled using solute temperatures in 16 copies of the original systems (*Figure 1—figure supplement 3*). After modeling missing residues, two simulations were started from a Down cryo-EM structure (PDBID: 6VXX; *Walls et al., 2020*) in the presence (gREST_Down) and absence of glycan (gREST_Down w/o glycan), and one from an one-Up structure (PDBID: 6VYB; *Walls et al., 2020*) (gREST_Up). MD simulations for longer than 15 µs, in total, were carried out for the solvated S-protein including about 655,000 atoms. Analyses of SD1/RBD hinge angle and the Cα root mean square deviation (RMSD) of each RBD upon fitting S2 provide simple measures of the RBD Down-to-Up transitions: the RMSD and hinge angle in Up take the values of about 20 Å and 150°, respectively, while the hinge angle in Down is about 116°. *Figure 1—figure supplement 4* indicates that RBD motions are drastically enhanced in all three simulations. Since gREST_SSCR is free from any pre-defined reaction coordinates, the Down-to-Up transitions happened not only in $RBD_A$ but also in $RBD_B$ or $RBD_C$. Even two-Up-like conformations were observed in some replicas, for instance, replica 8 in gREST_Down, replica 4 in gREST_Up, and replica 16 in gREST_Down w/o glycan (*Figure 1—figure supplement 4*).

**Table 1.** Molecular dynamics (MD) simulations of spike (S) protein performed in this study.

| Name | Model | Method | Simulations length |
| --- | --- | --- | --- |
| gREST_Down | Spike Down w/ glycans | gREST_SSCR | 500 ns × 16 replicas |
| gREST_Up | Spike Up w/ glycans | gREST_SSCR | 300 ns × 16 replicas |
| gREST_Down w/o glycan | Spike Down w/o glycans | gREST_SSCR | 150 ns × 16 replicas |
| Monomer_Down | RBD/SD1 monomer Down | cMD | 300 ns × 1 run |
| Monomer_Up | RBD/SD1 monomer Up | cMD | 300 ns × 2 runs |
| cMD_Down* | Spike Down w/ glycans | cMD | 1000 ns × 1 run |
| cMD_Up* | Spike Up w/ glycans | cMD | 1000 ns × 1 run |

*cMD_Down and cMD_Up are the same simulations as shown in our previous study *Mori et al., 2021*.

Despite the large-scale motions, intra-domain structures of three RBDs and NTDs were kept stable (*Figure 1—figure supplement 5*).

## Simulated structures connect cryo-EM ensembles

As of September 2021, over 300 cryo-EM structures of S-protein, including all Down, one-Up, two-Up, and three-Up conformations, were deposited in Protein Data Bank (*Supplementary file 1A*). Considering the trimeric nature of S-protein and its complex motions, the comparison with experimental structures is not trivial. For this purpose, we introduce (1) the nine-beads representation per protomer as used in the studies by Henderson et al. (*Gobeil et al., 2021*; *Henderson et al., 2020a*) (the 27-beads model of S-protein) (*Figure 1a* and *Supplementary file 1B*), (2) the rotation scheme to make the most significant RBD motion always happen in Chain$_A$ (*Figure 1—figure supplements 6 and 7*), and (3) the principal component analysis (PCA) on the 27-beads model upon fitting all the beads to reduce the essential dimensions in the conformational space. In *Figure 1b*, the first principal component (PC1) represents a symmetric Down-to-Up motion involving all three RBDs, while the second component (PC2) reveals an asymmetric motion of RBDs where only RBD$_A$ undergoes the Down-to-Up motion. The two lowest PCs cover about 85% of the conformational variations observed in the cryo-EM structures. In *Figure 1c*, the cryo-EM structures with distinct RBD conformations, Down, one-Up, two-Up, and three-Up, are found in different ensembles on the PC1-PC2 space (*Figure 1c*).

Next, we project the results of our previous 1 μs conventional MD (cMD) simulations (*Mori et al., 2021*) (cMD_Down and cMD_Up in *Table 1*) and the three gREST_SSCR simulations at 310 K on the same space (*Figure 1c*). The gREST_SSCR distributions at the temperature were obtained using multistate Bennett acceptance ratio (MBAR) (*Shirts and Chodera, 2008*) by utilizing all the replicas' trajectories. The structure distributions in cMD_Down and cMD_Up overlap with the corresponding Down and one-Up cryo-EM structures, while there is a big gap between the distributions on the PC1-PC2 space. Instead, gREST_SSCR could fill the gap effectively: gREST_Down samples from Down to one-Up, while gREST_Up covers one-Up and two-Up ensembles. Using the simulated distributions, we can investigate the inherent flexibility of S-protein, the transition pathways, and the intermediate structures. gREST_Down w/o glycan gives a wider conformational space than gREST_Down, suggesting that glycans on the surface of S-protein play key roles in the conformational stability.

To focus on the Down-to-Up motions in S-protein, we define the hinge and twist angles using the Cα atoms in RBD and SD1 (*Figure 2b*). The hinge angle directly describes the Down-to-Up transitions, while the twist angle explains side motions of RBD. Larger values of the hinge and twist angles signify the transition toward Up forms. The gREST_SSCR distributions at 310 K on the hinge-twist angle space (*Figure 2—figure supplement 1*) well overlap with those of the 891 protomers in the 300 cryo-EM structures, as we found on the PC1-PC2 space. Interestingly, the cMD simulation consisting of only a single monomer of SD1 and RBD in the Up form (Monomer_Up in *Table 1*) sampled Down, one-Up, and one-Open states within 300 ns and gave the structure ensemble close to the cryo-EM structure distributions and the gREST simulations containing a whole S-protein.

On the Hinge$_A$-Hinge$_B$ space, a second Down-to-Up motion is also identified for Chain$_B$ in gREST_Up (*Figure 2c*). We did not observe similar motions on the Hinge$_A$-Hinge$_C$ space, suggesting the order of Down-to-Up transitions toward two-Up. Taking together the ensemble distributions on the PC1-PC2 and the Hinge$_A$-Hinge$_B$ spaces, four major structures were identified: Down symmetric (Down$_{Sym}$: (Hinge$_A$, Hinge$_B$) = (114.9°, 112.5°)), one-Up (1U: (158.3°, 112.7°)), one-Up-open (1U$_O$: (163.3°, 108.9°)), and two-Up-like (2U$_L$: (150.6°, 148.6°)). They are superimposed to the cryo-EM structures having corresponding S-protein conformations in *Figure 2d*. Down$_{Sym}$ and 1U from our simulations are well aligned to the high-resolution cryo-EM structures of Down (PDB:6ZGE; *Wrobel et al., 2020* [RMSD: 3.7 Å]) and one-Up (PDB:6XKL (*Hsieh et al., 2020*) (RMSD: 3.6 Å)), respectively. Intriguingly, RBD$_A$ in 1U$_O$ is aligned to one of the RBDs in one of the three-Up structures (PDB:7DCC; *Zhang et al., 2021b* [RMSD: 5.4 Å]). Although 2$_{UL}$ from our simulations remains some interactions between RBD$_A$ and RBD$_B$, which is completely lost in two-Up cryo-EM structures, 2$_{UL}$ is aligned with those in one of the two-Up structures (PDB:6X2B; *Henderson et al., 2020a* [RMSD: 5.8 Å]). The overall comparison of the simulations' structures to cryo-EM points out the quality of the predicted structures.

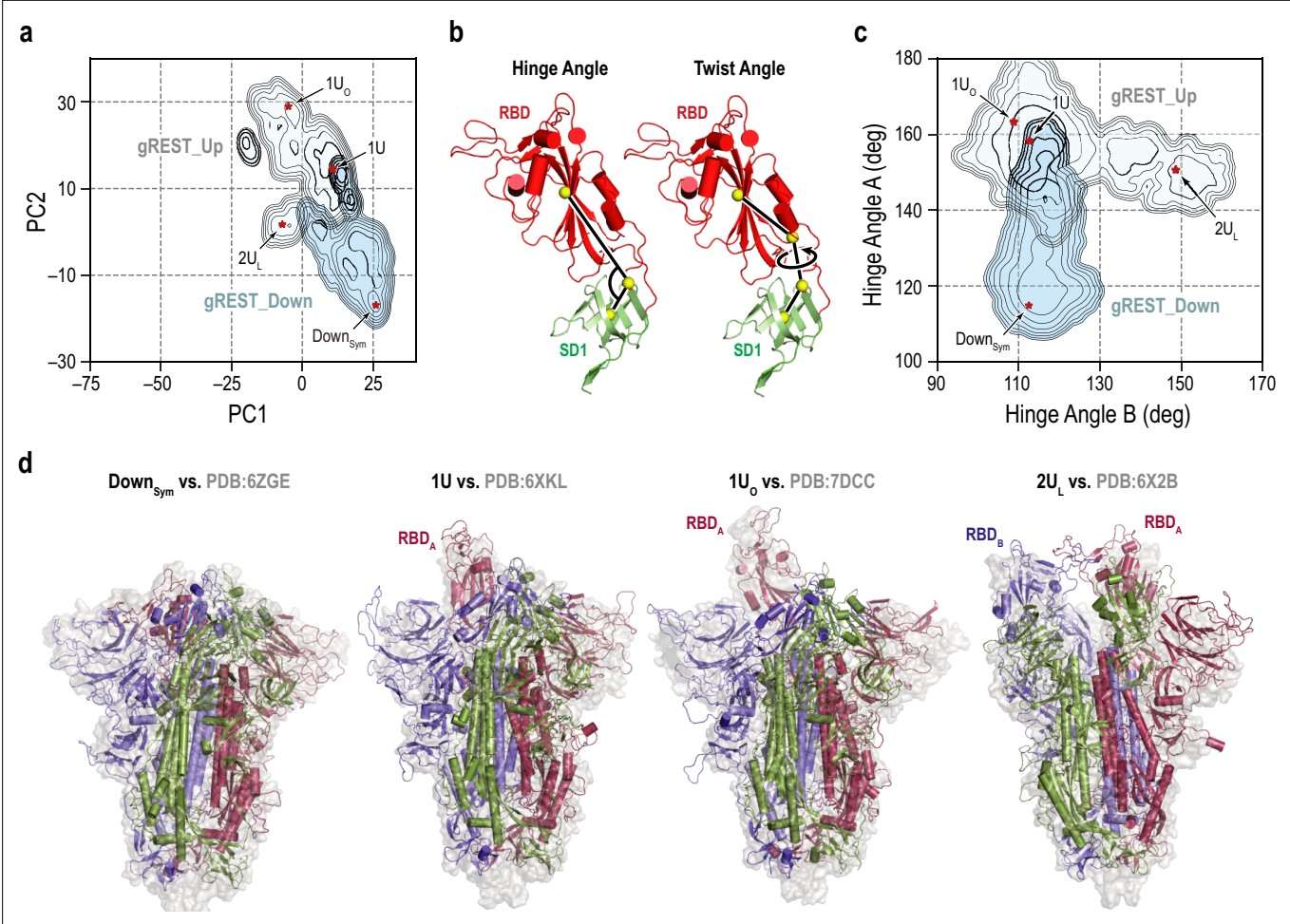

**Figure 2.** Representative receptor binding domain (RBD) conformations from molecular dynamics (MD) simulations vs. cryo-electron microscopy (cryo-EM) PDB structures. (**a**) An overlay of the two free-energy landscapes at 310 K along the PC1 and PC2 obtained from gREST_Down (light blue) and gREST_Up (light cyan) simulations. The red dots represent the positions of four representative RBD conformations: symmetric Down (Down$_{Sym}$), 1RBD Up (1U and 1U$_O$) and 2RBDs Up-like (2U$_L$) conformations. (**b**) Definitions of the hinge and twist angles representing the RBD conformations. The hinge angle is determined by three centers of masses (COMs, yellow spheres) of the core and top residues of SD1 (green, Cα atoms only) and the core residues of RBD (red, Cα atoms only). The twist angle is determined by the aforementioned COMs with an extra COMs of the bottom residues of RBD. (**c**) An overlay of two free-energy landscapes at 310 K along the hinge angles in RBD$_A$ and RBD$_B$ obtained from gREST_Down (light blue) and gREST_Up (light cyan) simulations. (**d**) Four representative conformations from MD simulations (cartoon representation) in comparison with cryo-EM structures (light gray surface). Chains A, B, and C in spike (S) protein are shown in red, blue, and green, respectively.

The online version of this article includes the following figure supplement(s) for figure 2:

**Figure supplement 1.** Free-energy landscape (FEL) along the hinge/twist angles in the gREST_SSCR simulations in comparisons with the conventional molecular dynamics (cMD) simulations for receptor binding domain (RBD)/SD1 monomer and cryo-electron microscopy (cryo-EM) structures.

## The accessibility of RBD in different conformations

To get insights into the contribution of each conformation to ACE2 and neutralizing antibodies (nAbs) binding, the accessibility of RBD is examined in terms of the solvent accessible surface area (SASA) (*Casalino et al., 2020*). *Figure 3a and b* and *Figure 3—figure supplements 1–5* show the per-residue SASA around the receptor binding motif (residues I410 to V510, referred to as RBM hereafter) calculated for Down$_{Sym}$, 1U, 1U$_O$ and 2U$_L$. RBM is rarely exposed in Down$_{Sym}$, where glycans at N165 and N343 largely shield RBM. In contrast, RBM SASA increases in 1U and 1U$_O$ compared to Down$_{Sym}$, wherein 1U$_O$ exhibits the utmost increase, suggesting its potential contribution to the receptor and antibody binding. The SASA increases only slightly, even decreases locally around residue C480, in 2U$_L$ due to the interactions between RBD$_A$ and RBD$_B$. This finding suggests that one-Up is the primary target of the receptor and antibody binding. Focusing on mutational residues of concern (K417, L452,

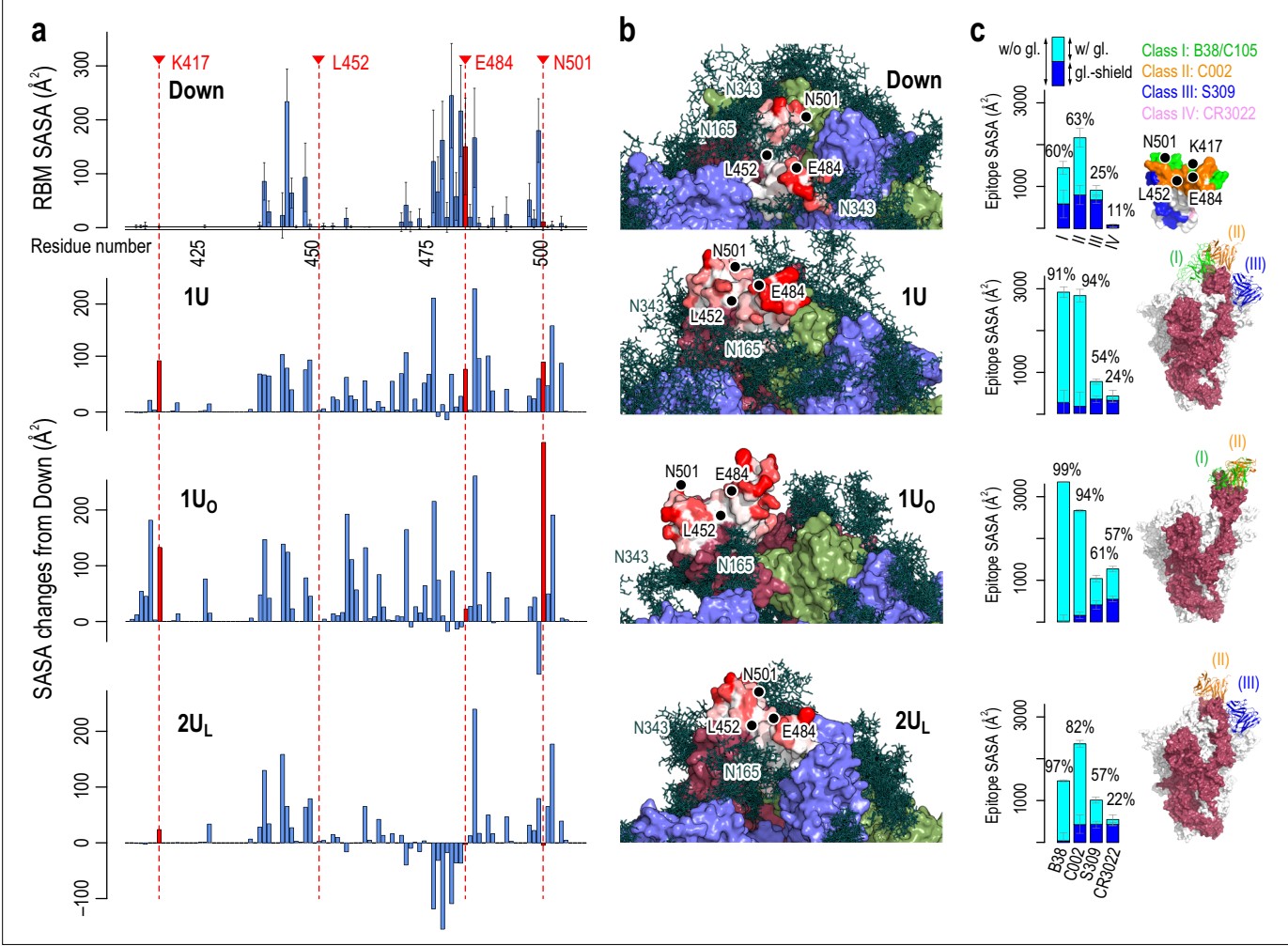

**Figure 3.** Accessibility of receptor binding motif (RBM). (**a**) Per-residue solvent accessible surface area (SASA) values of the RBM (residues 410–510) in Down conformation (top) and their changes in Up conformations (bottom three). SASA values were calculated using the probe radius of 7.2 Å. Four mutational residues, K417, L452, E484, and N501, are highlighted in red. (**b**) The surface representation of RBM SASA (white to red for 0–260 Å², the maximum value in Down, the values higher than this are truncated for consistent color scheme). Chains A, B, and C in the protein are shown in red, blue, and green surfaces, respectively, while a collection of glycans from 10 snapshots are shown in stick representation. (**c**) Epitope SASA and glycan shielding of four types of neutralized antibodies, B38 (Class I), C002 (Class II), S309 (Class III), and CR3022 (Class IV). Sum of SASA with glycans (cyan) and the glycan shield (blue) gives SASA without glycans. The ratio of the SASA with glycan over that without glycans is shown. The right-most column shows the putative interaction models with three classes of antibodies: Class III (S309, PDBID: 6WPT, blue), Class II (C002, PDBID: 7K8T, orange), and Class I (C105, PDBID: 6XCM, green).

The online version of this article includes the following figure supplement(s) for figure 3:

**Figure supplement 1.** Glycan shield of spike (**S**) protein.

**Figure supplement 2.** Accessibility of receptor binding motif (RBM).

**Figure supplement 3.** Glycan effect on the accessibility of receptor binding motif (RBM).

**Figure supplement 4.** Putative interaction models with antibodies.

**Figure supplement 5.** Comparison of Up structures from molecular dynamics (MD) simulations with cryo-electron microscopy (cryo-EM) structures.

E484, and N501 *Harvey et al., 2021*), K417 and N501 are accessible only in one-Up conformations. The mutations of these residues could directly affect the binding with the receptor and antibody. On the other hand, E484 is accessible in both Down and Up, inferring a widespread effect of E484 mutations. L452 is not accessible in any conformations and hence is expected to rarely affect the binding with either receptor or antibody in wild type (*Valdes-Balbin et al., 2021*).

To examine antibody preferences in Up conformations, we also calculate SASA for epitope regions that are recognized by four classes of nAbs (Class I: binds to Up for blocking ACE2, Class II: binds to Up and Down for blocking ACE2, Class III: binds outside RBM but recognize Up and Down, Class IV: binds Up without blocking ACE2) (*Harvey et al., 2021*). The epitope regions of Up conformations are generally less shielded by glycans compared to Down (at most 99% de-shielded, *Figure 3c*). Notably, the epitope SASAs of Class I and II antibodies in 1U and $1U_O$ are significantly large. The C105 (Class I) and C002 (Class II) antibodies in cryo-EM structures are well aligned to 1U and $1U_O$ without steric crash (*Figure 3c* and see *Figure 3—figure supplement 4* for detail). These results suggest that Class I and II antibodies preferentially bind to these two conformations. The corresponding SASA values are small in $2U_L$ due to the interaction between $RBD_A$ and $RBD_B$, leading the Class I antibody difficult to align without steric crash. Note that there are differences between $2U_L$ and the full two-Up cryo-EM structures. The observation on SASA values in $2U_L$ is applicable to the state just before reaching the full two-Up structures. In any Up conformations, the epitope SASAs of Class III antibody is comparable to that of Down, while the region is less shielded by glycans. The S309 (Class III) antibody in cryo-EM structures is still well aligned to 1U and $2U_L$, rationalizing that the Class III antibodies recognize 'glyco-epitopes' (*Pinto et al., 2020*). Note that the epitope region of the Class IV antibody is slightly exposed only in $1U_O$. Further opening of RBD may allow the binding of the Class IV antibody, such as CR3022, as suggested in the previous computational study (*Zimmerman et al., 2021*). All these results suggest the preferential binding of different antibodies depending on the RBD conformation. Notably, the Up structures from MD simulations are reasonably aligned to those of cryo-EM structures with antibodies (*Figure 3—figure supplement 5*). This observation supports an idea of conformer selection regime for antibody bindings. As the S-protein RBD is inherently flexible, potential conformations for antibody bindings are likely programmed.

## Simulated structure ensembles align with smFRET experimental results

Using smFRET, Lu et al. examined the conformational dynamics of S-protein in the presence or absence of its receptor, hACE2 (*Lu et al., 2020*). They characterized four structural ensembles including two types of Down (major and minor), at least one intermediate and one-Up conformations, suggesting the inherent flexibility of the RBD region regardless of the receptor binding. However, to date, there is no structural information that describes the observed intermediate states in the smFRET experiment. In addition, despite the abundance of cryo-EM structures, they are unable to solely explain smFRET results as shown in *Figure 4—figure supplement 4d*. To unravel such inaccessible structural information of the intermediate states, we first classify each trajectory at 310 K using k-means clustering and re-clustering guided by the hinge and twist angles distributions (*Figure 4—figure supplements 1–3*, *Supplementary file 1C*). The 13 micro-clusters were identified from gREST_Down (*Figure 4—figure supplement 1*) including two clusters representing the symmetric Down (Down_sym: $D1_{Sym}$ and $D2_{Sym}$), two clusters of asymmetric Down with differences in RBDs twist angle distribution ($Down_{Asym}$: $D1_{Asym}$ and $D2_{Asym}$), three clusters with a slight increase in the RBD hinge angle around 120°, (Int1: I1a, I1b, and I1c), three clusters with one RBD hinge angle around 130° (Int2: I2a, I2b, and I2c), two clusters with hinge angle around 140° (Int3: I3a and I3b), and one cluster with an Up-like conformation ($1U_L$). Similarly, from gREST_Up, the 13 clusters were identified including 10 clusters representing one-Up conformation (1U: 1Ua-1Uj), two clusters with two-Up-like conformations ($2U_L$: $2Ua_L$ and $2Ub_L$) and one-Up open cluster ($1U_O$). Then we compute the distance between residues 425–431 in $RBD_A$ and 554–561 in $SD1_A$ to correlate the simulated structure ensembles with the smFRET intensity reported by Lu et al., 2020. By combining the distance distributions from gREST_Down and gREST_Up, we obtain five main conformational ensembles (*Figure 4—figure supplement 4*): Down symmetric ($Down_{Sym}$), Down like ($Down_{Like}$: $Down_{Asym}$ and Int1), two intermediates (Int2 and Int3), and Up (1Up). The smFRET distance alone cannot distinguish various Up conformations including 1U, $1U_O$, and $2U_L$ and thus we refer to them as 1Up. $Down_{Sym}$ and $Down_{Like}$ give distributions in the range of 30–35 Å, while 1Up shows around a median distance of 47 Å. The distributions of Int2 and Int3 have median distances of around 38 and 40 Å, respectively. Note that Int3 has a large distance distribution that overlaps with Int2 and both intermediates might be indistinguishable in the smFRET experiment. Collectively our simulations show the formation of four main conformations $Down_{Sym}$, $Down_{Like}$, Intermediate (Int2 and Int3), and 1Up that correspond to the smFRET intensity of 0.8, 0.5, 0.3, and 0.1 in the absence of hACE2 (*Lu et al., 2020*). Each ensemble is also characterized with hinge angle distributions that align

in order of smFRET distances: $Down_{Sym}$: Hinge < 120°, $Down_{Like}$: Hinge < 130°, I2: 120° < Hinge < 140°, I3: Hinge ~140°, 1Up: 140° < Hinge < 160°.

## Transition pathways and transient interactions stabilizing the intermediate structures

We next focus on molecular mechanisms underlying the Down-to-1Up transitions in terms of the correlated motions of the hinge and twist angles in the three RBDs (*Figure 4—figure supplement 5*). Note that such correlated motions are hardly obtained using targeted MD simulations (*Figure 4—figure supplement 6*) or the MD simulations enhanced with pre-defined reaction coordinates (*Sztain et al., 2021*; *Brotzakis et al., 2021*). *Figure 4—figure supplement 5* indicates that the transition pathway from Down-to-1Up conformation occurs via four main states including two intermediates. First, the transition is initiated from flexible Down structures ($Down_{Sym}$ and $Down_{Like}$), where three RBDs show high flexibilities reflected by a wide range of twist angle between 10° and 90° while $RBD_A$ maintains a hinge angle <125°. The increase in $RBD_A$ hinge angle to 130° in I2a is accompanied by the reduction in other RBDs' flexibility. It is reflected by smaller changes in twist and hinge angles for the chains B and C. Later, two main pathways are identified through I3a or I3b (hinge 140°), however, comparisons of all five free-energy maps as well as the projection on PC1-PC2 free-energy surface suggest that I3b is an off-target intermediate. In I3a, both hinge and twist angles are increased in $RBD_A$, accompanied by a slight reduction in $RBD_C$ hinge angle. Both I2a and I3a represent stable basins along the free-energy landscapes (FELs) along the PC1-PC2 and hinge-twist angles, emphasizing their role in mediating the transition (*Figure 4—figure supplements 5 and 11*). Lastly, $1U_L$ conformation is formed with hinge angle >150° with a concomitant increase in its twist angle. In general, $Hinge_A$ and $Twist_A$ are highly correlated (cc: 0.94) with each other (*Figure 4—figure supplement 5c*), while $Hinge_A$ and $Hinge_B$ show no correlations (cc: 0.02). The correlations between $Hinge_A$ and $Hinge_C$/$Twist_C$ exist to some extent (cc: –0.68 and 0.30).

To further analyze the changes throughout the transition pathway, the analysis of contact (*Figure 4—figure supplement 7*) and hydrogen bond (HB) (*Figure 4—figure supplement 8*) were carried out for each cluster observed in gREST_Down and gREST_Up as we did in the previous study (*Mori et al., 2021*). In *Figure 4*, several key contacts and HBs are highlighted to see drastic changes of the interactions in the Down-to-1Up transition. The glycans at $N343_B$ and $N234_B$ switch the interactions with $RBD_A$ along the transition from Down to one-Up-like. In the intermediates, the glycan at $N343_B$ changes its contact partners by inserting underneath $RBD_A$. Concurrently, the formation of salt-bridge interactions between $RBD_A$ and $RBD_C$, for instance, $R408_A$-$D405_C$ and $R408_A$-$D406_C$, lift up $RBD_A$ from its Down to intermediate structures. Finally, $N234_B$ inserts into the newly formed cavity in $1U_L$ between $RBD_A$ and $RBD_B$, forming new contacts with $D428_A$ and $T430_A$. The glycan at $N165_B$ is in contact with $RBD_A$ throughout the transition, likely acting as a main barrier.

The formation of two Up-like conformation is characterized by the increase of hinge/twist angle in $RBD_B$, as shown in *Figure 1—figure supplement 7c*. Free energy landscape (FEL) on $Hinge_A$-$Twist_A$ angles shows a similar transition as observed in $RBD_A$ in gREST_Down (*Figure 1—figure supplements 7c and 6d*). The relative RBD motions during the 1Up-to-2Up transition is characterized using the hinge angles (*Figure 5—figure supplement 1a*). The increase of hinge angle in $RBD_B$ is accompanied with a slight decrease of hinge angle in $RBD_A$ in $2Ua_L$ and subsequently in $2Ub_L$. Note that both clusters might represent intermediate states toward the full 2Up structure, in which the increase of $RBD_A$ hinge angle is regained. The latter decrease likely relates with the increases of the contacts and HBs between $RBD_A$ and $RBD_C$, for instance, $K378_A$-$E484_C$, $Y369_A$-$N487_C$, and $D427_A$-$Y505_C$, from 1Ua, the top populated cluster in 1U, to $2U_L$ structures (*Figure 5* and *Figure 4—figure supplements 7 and 8*). The numbers of contacts and HBs in two-Up-like structures are much less than those in 1Ua, suggesting that a drastic reduction of the inter-domain interactions between $RBD_A$ and $RBD_B$ is required to form a full two-Up conformations. No specific protein-glycan interactions are found to support the 1Up-to-2Up transition, while more contacts between $RBD_B$ and glycan at $N165_C$ are observed in the two-Up-like conformations, emphasizing its role as a bottleneck of the transition.

We also examine the effect of glycans on the Down-to-1Up transitions. *Figure 5—figure supplement 1b* shows that the distributions of gREST_Down w/o glycan on the $Hinge_A$-$Hinge_B$ space become much wider, suggesting that S-protein is more flexible without glycans. In addition, Down conformations are more diverse and asymmetric as one RBD has larger hinge angle distributions than the

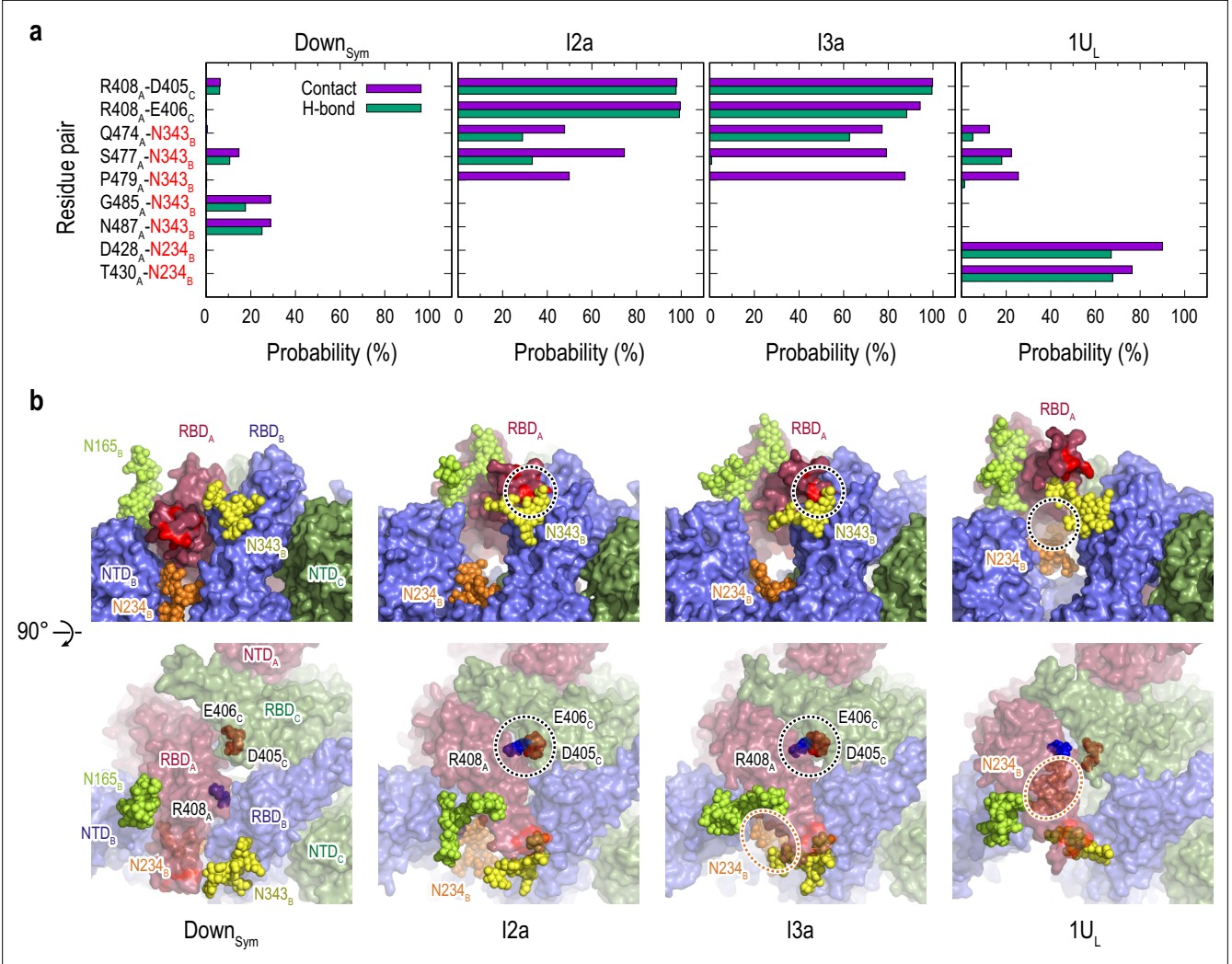

**Figure 4.** Protein-protein and protein-glycan interactions critical for Down-to-Up transition. (**a**) Probability of finding the hydrogen bond (green) and contact (purple) pairs between protein residues or protein-glycans that markedly change along the transition pathway (Down$_{Sym}$, I2a, I3a, and 1U$_L$). All hydrogen bond (probability of finding of >50%) and contact pairs (probability of finding of >70%) are shown in *Figure 4—figure supplements 7 and 8*, respectively. (**b**) Typical snapshots of the protein-protein and protein-glycan interactions along the transition pathway. Chains A, B, and C in the protein are shown in red, blue, and green surfaces, respectively. Glycans at N165, N234, and N343 are shown with spheres in lime, orange, and yellow color, respectively. The transient N343$_B$-RBD$_A$ contact is highlighted in red surface (top). The salt-bridges formed by R408$_A$ (blue), E406$_C$ (brown), and D405$_C$ (red) are also highlighted with black dashed circles (top and bottom), while the location of N234 glycan is highlighted with orange dashed circles.

The online version of this article includes the following video and figure supplement(s) for figure 4:

**Figure supplement 1.** Clustering for the conformations obtained from the gREST_Down simulation.

**Figure supplement 2.** Clustering for the conformations obtained from the gREST_Up simulation.

**Figure supplement 3.** Clustering for the conformations obtained from the gREST_Down w/o glycan simulation.

**Figure supplement 4.** Simulated single molecule fluorescence resonance energy transfer (smFRET) distance using the gREST_SSCR trajectory data.

**Figure supplement 5.** Transition pathway from Down to 1Up in the gREST_Down simulation.

**Figure supplement 6.** Comparison between targeted molecular dynamics (TMD) and gREST_SSCR simulations.

**Figure supplement 7.** Contact analysis for the main clusters in gREST_Down and gREST_Up simulations.

**Figure supplement 8.** Hydrogen bond analysis for the main clusters in gREST_Down and gREST_Up simulations.

**Figure supplement 9.** Relationship between the sideway motion of RBD$_B$ and intrusion of the glycan at N343$_B$.

**Figure supplement 10.** Glycan interaction sites in various Up conformations.

**Figure supplement 11.** Stability of I2a and I3a along transition pathway.

*Figure 4 continued on next page*

*Figure 4 continued*

**Figure supplement 12.** Intermediate structures align with cryo-electron microscopy (cryo-EM) structures.

**Figure 4—video 1.** Protein-glycans interactions along Down-to-1Up transition pathway.

https://elifesciences.org/articles/75720/figures#fig4video1

others. The transition mechanisms seem to be the same as those with glycans: RBD$_A$ and RBD$_C$ form the transient interactions to support the Down-to-1Up transition. However, the two-Up-like structure distribution is directly connected with Down, suggesting that structural integrity of S-protein is lost without glycans.

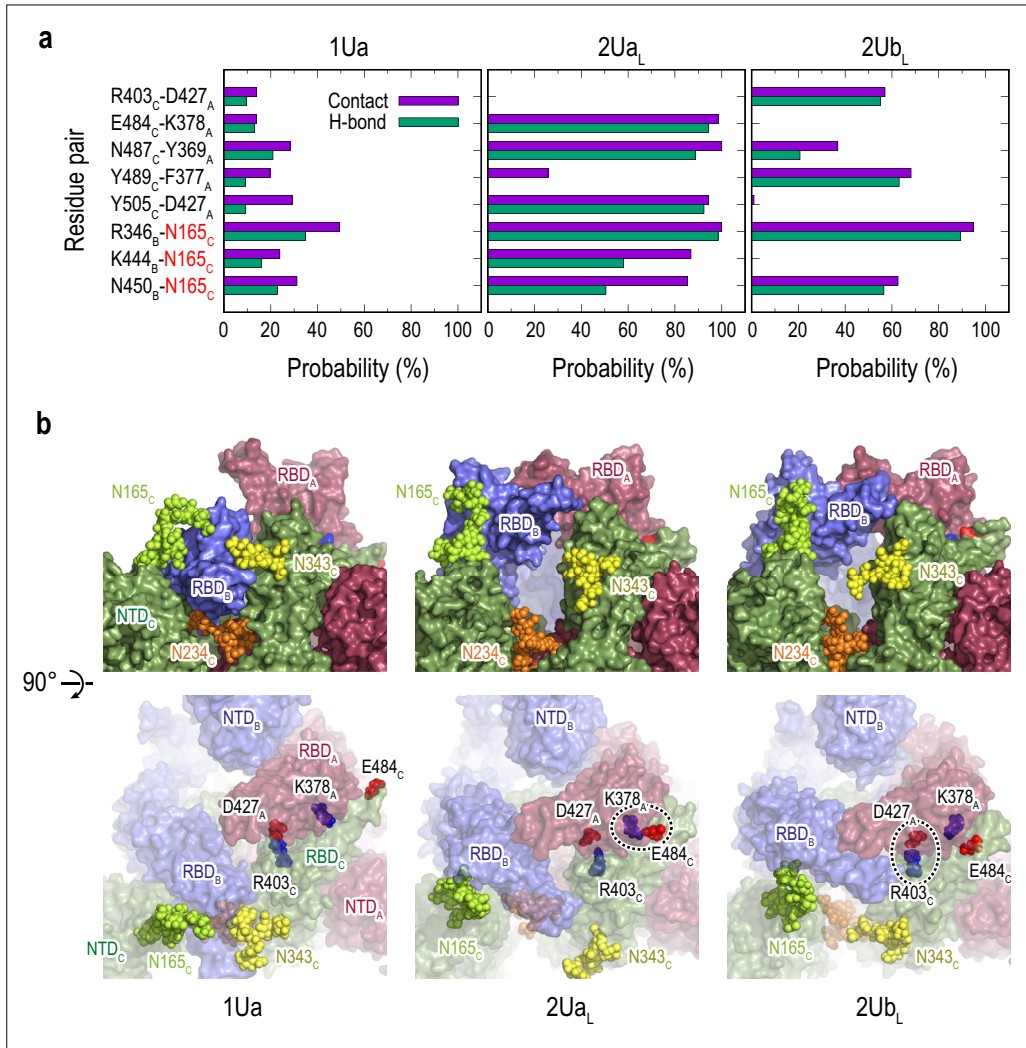

**Figure 5.** Selected protein-protein and protein-glycan interactions along the 1Up-to-2Up transition. (**a**) Probability of finding the hydrogen bond (green) and contact (purple) pairs throughout the transition pathway (1Ua, 2Ua$_L$, and 2Ub$_L$). Where the conformational transition of RBD$_B$ induces RBD$_A$/RBD$_C$ interactions. All hydrogen bonds and contact pairs are shown in *Figure 4—figure supplements 7 and 8* respectively. (**b**) Typical snapshots of the protein-protein and protein-glycan interactions along RBD$_B$ transition pathway. Chains A, B, and C in the protein are shown in red, blue, and green surfaces, respectively. Glycans at N165, N234 and N343 are shown with spheres in lime, orange, and yellow colors, respectively. Positive and negatively charged residues are shown as blue and red spheres, respectively. The transient salt-bridges formed by K378$_A$/E484$_C$ and D427$_A$/R403$_C$ are also highlighted with black dashed circles.

The online version of this article includes the following figure supplement(s) for figure 5:

**Figure supplement 1.** Free-energy landscape (FEL) in the gREST_Up and gREST_Down w/o glycan simulations.

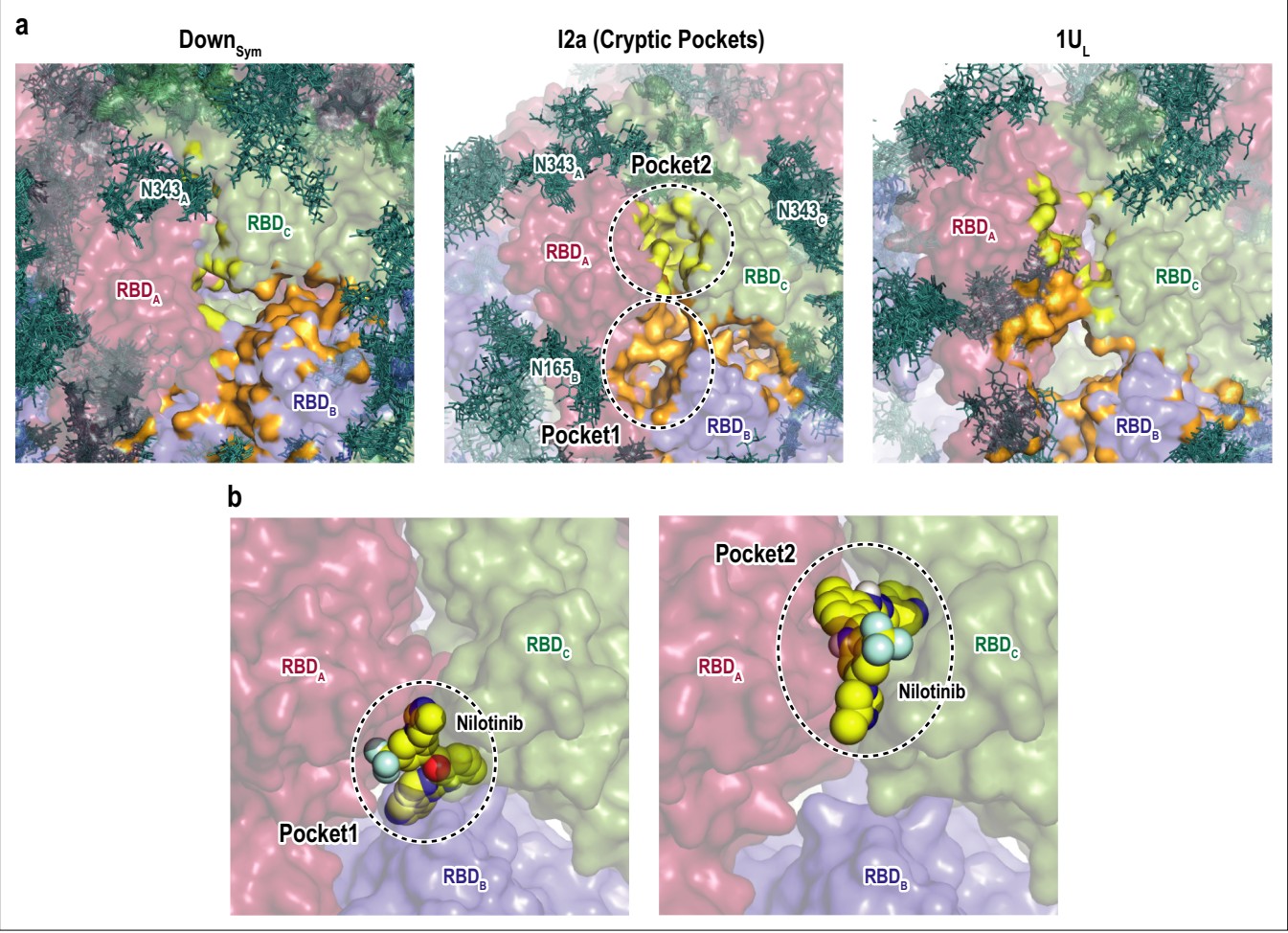

**Figure 6.** Druggable cryptic pockets in the transition intermediates. (**a**) Snapshots of receptor binding domain (RBD) interface in Down symmetric (Down_Sym), Intermediate 2a (I2a), and 1Up-like (1U_L) conformations. Chains A, B, and C in the protein are shown in red, blue, and green surfaces, respectively, while a collection of glycans from 10 snapshots are shown in stick representation. The cryptic pockets predicted for I2a using P2Rank software are shown in orange (Pocket1) and yellow (Pocket2), respectively. These pockets disappear in both Down_Sym and 1U_L. (**b**) Nilotinib docked poses (top and third ranked) to two cryptic pockets in I2a by Autodock Vina. The pockets are highlighted with black dashed circles and the nilotinib is shown in sphere representation in yellow.

The online version of this article includes the following figure supplement(s) for figure 6:

**Figure supplement 1.** Cryptic pockets and ligand binding in receptor binding domain (RBD).

## Searching for cryptic binding pockets in the intermediate structures

We applied a machine learning-based algorithm (P2Rank) (*Krivák and Hoksza, 2018*) to search for the formation of druggable pockets in the intermediate structures (*Figure 6—figure supplement 1a*). The same search was also carried out for Down_Sym and 1U_L for comparison. *Figure 6a* shows the formation of two cryptic pockets (pocket1 and pocket2) at the interface of RBDs in one of the intermediates, which is not observed in Down or one-Up. *Supplementary file 1D* lists these predicted pockets showing relatively high scores in all the intermediates. To test the druggability of these two pockets, we performed virtual screening of FDA approved drugs from ZINC database (*Irwin and Shoichet, 2005*), where we docked 2115 molecules to the RBD interfaces in the intermediates (following the procedure sketched in *Figure 6—figure supplement 1c*). *Supplementary file 1E* shows a list of the index (ID) and binding energies of top-ranked molecules. This includes few anticancer drugs (irinotecan, nilotinib, and ponatinib), antimigraine drugs (ergot alkaloids), and antiviral-drug (glecaprevir). Intriguingly, the previous experimental study has shown that nilotinib affects the SARS-CoV-2 infectivity, although the mechanism remains unknown (*Cagno et al., 2021*). *Figure 6c* and *Figure 6—figure supplement 1d* show the representative binding poses of nilotinib in the three intermediates.

The top-ranked drugs tend to bind to either pocket1 or pocket2 in all first nine binding modes with high binding affinity (*Supplementary file 1F*). These results serve the two cryptic pockets as potential targets to stabilize the intermediate structures and prevent the formation of one-Up conformation responsible for the viral entry.

## Discussion

### Sampling the conformational space of S-protein

Due to the importance of the conformational changes of S-protein in the infection mechanisms and the rational design of antiviral drugs or antibodies, many extensive simulation studies on S-protein have been carried out since the start of the pandemic. The latest challenges involve the use of a million of distributed computer resources, Folding@home, to realize MD simulation of the SARS-CoV-2 proteome for 0.1 s, in total. An alternative approach is to incorporate experimental observation to effectively sample different states. Brotzakis and co-workers recently applied the cryo-EM metain-ference method to determine the opening pathway and intermediates based on the experimental density map (*Brotzakis et al., 2021*). Yet another approach is to use enhanced sampling methods. Sztain and co-workers performed the weighted ensemble simulations, collecting 130 μs trajectories, to characterize the opening pathway (*Sztain et al., 2021*). Pang et al. employed two-dimensional umbrella sampling to characterize the transition pathway and the role of glycans (*Pang et al., 2021*). The gREST_SSCR simulations are free from reaction coordinates and bias potentials that were used in many previous enhanced sampling simulations of S-protein. By using Fugaku supercomputer as well as GENESIS program (ver2.0) designed for achieving high scalability on that system (*Kobayashi et al., 2017*; *Jung et al., 2021*), the simulations explored a wide conformational space covering Down, one-Up, one-Open, and two-Up-like structures. The obtained structure ensembles overall agree with the cryo-EM structure distributions and smFRET data, suggesting the existence of the inherent flexibility of S-protein structures without its ligand, such as hACE2 or nAbs. Comparison to cryo-EM and smFRET experiment also indicates the quality of the predicted structures including intermediate states. In fact, comparison of the intermediate structures I2a and I3a with the D614G spike variant cryo-EM structure (PDB:7KRS) (*Zhang et al., 2021a*) shows a good agreement, with Cα RMSD of 4.9 and 5.1 Å, respectively (*Figure 4—figure supplement 12*).

Besides the enhancement of conformational sampling by gREST_SSCR, there are several remaining issues in the simulations. Within our computational time, one single simulation cannot cover a whole FEL containing all the important structures of S-protein. Three-Up structures, which were found with cryo-EM, for instance, PDB:7DCC (*Zhang et al., 2021b*), were not observed in the current simulations. As we discuss in the next section, glycans on the surface of S-protein seem to have dual roles in the transitions, either helping or hindering the Down-to-Up transitions, which makes conformational sampling via MD simulations more difficult. However, conformational sampling using gREST_SSCR is much better than cMD without any pre-defined reaction coordinates and/or bias potentials, providing sufficient structural information to investigate the inherent flexibility of S-protein and the transition mechanisms.

### Role of flexible Down structures in the Down-to-Up transitions

In this study, we observed the inherent flexibility of S-protein in absence of its ligand, exploring Down, one-Up, one-Open, and two-Up-like structures at 310 K. The flexible nature of RBD forming multiple Down and Up conformations was also demonstrated in the cryo-EM study at physiological pH by *Pramanick et al., 2021*. The trimeric RBDs in S-protein are intrinsically flexible even in Down, as their interfaces are electrostatically repulsive (*Mori et al., 2021*). Not only symmetric structures but also anti-symmetric structures are also observed in Down, suggesting the existence of 'flexible Down structure' (*Figure 4—figure supplement 9*). This flexibility can initiate the Down-to-1Up transition, by allowing the transient reduction of inter-domain interactions in Down$_{Sym}$ and allowing the insertion of a glycan at N343$_B$ underneath RBD$_A$ (*Figure 4—figure supplement 10*). This is consistent with the mobile RBD conformations in Down observed experimentally by *Ke et al., 2020* and *Wrobel et al., 2020*.

Following the flexible Down structures, our simulations elucidate molecular mechanisms underlying the Down-to-1Up transition involving RBD$_A$ and the 1Up-to-2Up transitions involving RBD$_A$ and

RBD$_B$. This sequence of events can be explained via the changes of inter-domain interactions as well as protein-glycan interactions. As RBD$_A$-RBD$_C$ interaction becomes stronger in the Down-to-1Up transition (*Figure 4*, *Figure 4—figure supplements 7 and 8*), RBD$_B$ becomes more mobile for the next 1Up-to-2Up transition. Among the inter-domain interactions, we pointed out the importance of transient salt-bridges and HBs between R408$_A$-D405$_C$ and R408$_A$-D406$_C$ for stabilizing the intermediates (*Figure 4* and *Figure 4—figure supplement 8*). The salt-bridge between R408$_A$ and D405$_C$ was also observed in the weighted ensemble simulations reported by *Sztain et al., 2021*. This scenario is mediated by protein-glycan interactions, in particular, those involving three glycans at N165$_B$, N234$_B$, and N343$_B$ as pointed out in previous studies (*Choi et al., 2021*; *Casalino et al., 2020*; *Mori et al., 2021*). In the current study, we highlighted the dual roles of these three glycans: both to stabilize one of the states and to serve the driving force toward the other state. The glycan at N165 stabilizes Down, representing a barrier for the Down-to-1Up transition, while the glycan at N343$_B$ drives the transition. Finally, the glycan at N234$_B$ stabilizes one-Up-like conformation. This picture coincides with the experimental results by *Henderson et al., 2020b*, where the population of one-Up is drastically reduced by the glycan deletions at N234 but increases by those at N165. As suggested previously (*Casalino et al., 2020*), the glycans at N165 may also stabilize one-Up conformation but as a minor role because it can adapt various orientations to that conformation (*Figure 4—figure supplement 10*). The role of glycan at N343 is noteworthy in that it helps the position of RBD lift up from Down and supports it throughout the intermediates. This well explains the 20-fold reduction of the infectivity upon N343Q mutation observed by *Li et al., 2020*. Recently, a similar role of the glycan at N343 has been proposed by *Sztain et al., 2021*. Mutational studies of charged residues at the RBD interface (e.g. R408, D405, and D427) could validate the proposed role of the salt-bridge formation in the transition.

## Implication for vaccine and drug developments

As a consequence of its central role, S-protein has been early identified as potential target for drug repurposing in order to block viral entry. Since the drug repurposing studies (*Han et al., 2021*; *Bakowski et al., 2021*; *Deganutti et al., 2020*) mostly target Down or Up from cryo-EM structures, druggable cryptic pockets identified in the intermediates would introduce unprecedented drug targets (*Brotzakis et al., 2021*). Two cryptic pockets at the RBD interface are identified in the highly populated intermediates along the Down-to-1Up transition. Both I2a and I3a show high stability on the FELs on PC1-PC2 and hinge-hinge angles as shown in *Figure 4—figure supplement 11*. Notably, allosteric sites in intermediate structures were previously used for drug design (*Lu et al., 2021*; *Shukla et al., 2014*). From our virtual screening of FDA approved molecules, these pockets are druggable and accommodates several small molecules including irinotecan, ergotamine, nilotinib, and ponatinib. For example, nilotinib was shown to reduce SARS-CoV-2 infection by ~50% (*Cagno et al., 2021*), likely via blocking spike interactions with ACE2 (*Tsegay et al., 2021*). Notably, top-ranked molecules (*Supplementary file 1E*) were also predicted to bind RBD in the previous virtual screening studies (*Deganutti et al., 2020*; *Murugan et al., 2020*). If these bindings happen, the bound molecules at RBD interface could potentially stabilize the intermediate states and reduces the population of Up conformation, to block ACE2 binding. To shift the conformational equilibrium of S-protein toward the inaccessible Down state for blocking ACE2 and subsequent membrane fusion has been a focus of the recent challenges. Indeed, there have been some reports altering the conformational dynamics of S-protein either via site-specific mutations, disulfide bonds, and binding to small molecule (*Hsieh et al., 2020*; *Juraszek et al., 2021*; *Henderson et al., 2020a*; *Edwards et al., 2021*; *McCallum et al., 2020*; *Toelzer et al., 2020*). Despite the promising outcome from shifting S-protein equilibrium, the high rate of mutations in spike might hinder its use as drug and antibodies target or even vaccine development. Indeed, continuous assessment of cryptic and epitope regions in spike variants is necessary.

The understanding of antibody responses is critical for vaccines and nAb developments. The large exposure of RBD$_A$ in Up forms is shown in the current study, which is consistent with several previous studies (*Grant et al., 2020*; *Casalino et al., 2020*; *Sikora et al., 2021*). In consistent with the conformation-dependent SASA variation reported in the previous cryo-EM study (*Pramanick et al., 2021*), we found that the accessibilities of RBM and the antibody epitopes, including mutational sites, depend sensitively on the RBD conformation. The result suggests that each of the Up conformations (1U, 1U$_O$, and 2U$_L$) tends to bind with distinct classes of antibodies. From our

inspection, $1U_O$, which largely exposes RBM but the antibody epitopes of only Class I and II, is the potentially active conformation for both the ACE2 binding and evading from the antibody attack. Class I and II epitope regions include K417, E484, and N501, and their mutations could effectively enhance the infection either by enhancing the ACE2 binding (N501Y) (*Harvey et al., 2021*) or reducing the antibody binding (K417N or E484K) (*Harvey et al., 2021*). Intriguingly, HB analysis shows that K417 and E484 respectively contribute to stabilize the intermediates likely enhancing the population of one-Up conformations. Note that L452 shows a little accessibility (*Figure 3*) in contrast with the reported severe mutational effect to an antibody recognition (*Harvey et al., 2021*). The similar observation was also reported previously (*Valdes-Balbin et al., 2021*), while the recent simulation shows the possible interaction between L452 and an antibody (*Cao et al., 2021*). A recent comparison of L452R cryo-EM structure with wild type suggested that R452 sterically hinders the nAbs bindings without affecting ACE2 binding (*McCallum et al., 2021*). Finally, the Up conformations are reasonably aligned with the antibody-bound S-protein cryo-EM structures (*Figure 3—figure supplement 5*). Hence, the antibody bindings of S-protein can be explained based on the conformer selection mechanism (*Csermely et al., 2010*). The exploration of conformational diversity of S-protein together with binding free-energy calculations followed by docking simulation would provide valuable structural insights with possible antibody bindings (*Ren et al., 2021*).

Finally, we highlight the key differences between our simulations and other computational works on S-protein. Current work includes at least two unique features based on the simulation and analysis methods. First, our conformational sampling method, gREST, could avoid the use of pre-defined reaction coordinates and/or bias potentials along the coordinates. Therefore, the gREST simulation is able to generate unexpected structures or structural changes, which are free from the starting cryo-EM structures and the reaction coordinates in biased MD simulations. Two-Up-like structures, $Down_{Asym}$, or $1U_O$ were not expected even by us before the extensive gREST simulations and analysis. Since these unexpected structures involve novel molecular interactions, it is necessary to detect which interactions can stabilize the intermediate structures and to examine how the interactions are changed along the transition pathways. Therefore, we analyzed the HBs and contacts in each intermediate, systematically, and quantitatively (the second key feature). The changes of the atomistic interactions involving side chains and glycans could characterize the conformational transition pathways in atomic detail without using simplified reaction coordinates. To understand the effect of mutations in the conformational changes toward the active Up form, the difference of intrinsic flexibility between wild type and mutant S-protein is one of the most important features to be understood. The current simulation and analysis methods are applicable to new variants for giving atomistic pictures of conformational flexibility and transitions of spike proteins, which are both important for understanding the essential characters of new S-protein variants.

## Materials and methods
### gREST_SSCR simulations
The initial structures for S-protein head regions (residues 28–1135) were prepared based on the cryo-EM structures PDB:6VXX and PDB:6VYB for the Down and Up states, respectively (*Walls et al., 2020*). Eighteen N-glycans and one O-glycan were added per protomer as suggested in the mass spectrometry experiments and previous computational models (*Watanabe et al., 2020*; *Woo et al., 2020*). A full list of included glycans is shown in *Figure 1—figure supplement 2*. CHARMM-GUI (*Lee et al., 2016*) was used to prepare the final model including the glycans, ions (0.15 M NaCl), and water molecules. Three gREST_SSCR simulations were performed in which two started from Down in the presence (500 ns) and absence of glycans (150 ns), and one from Up (300 ns). Eight pairs of charged residues per protomer at the RBD interface were selected as the solute region for gREST. The total number of atoms in solute was 870. All simulations were performed using 16 replicas covering the solute temperature range from 310 to 545 K while maintaining solvent temperature at 310.15 K. All simulations were performed using the new version of GENESIS MD software that was optimized on Fugaku (*Kobayashi et al., 2017*; *Jung et al., 2021*). Further detailed information is given in Supporting Information.

## Comparison between cryo-EM structures and MD simulations using PCA

We used cryo-EM structures provided by 'spike protein and spike receptors' in Protein Data Bank (http://www.rcsb.org/covid19, deposited date 2020/02/04 to 2021/09/09, released by 09/22) in this study. Only structures where the number of residues is greater than 700 and the last residue ≥ 1122 for each protomer were selected in the analysis; 891 protomers of 300 structures and 289 trimeric structures from PDB meet the criteria (*Supplementary file 1A*, *Table 1*). We adopt a method representing the structure with nine beads per protomer as used in the previous work of *Gobeil et al., 2021*; *Henderson et al., 2020a*. Unlike their study, 27 beads consisting of three chains are used here. This coarse-grained model consists of two beads for RBD, three beads for NTD, one bead for SD1 and SD2, and two beads for the S2 region (CD and S2-b) (see *Supplementary file 1B*). We first executed PCA (*Yang et al., 2009*) of each protomer using the selected cryo-EM structures after converting the nine-beads model. Up/Down for each protomer was determined by the value of projection against the first PC vector. By using the information, the rotation scheme was applied to the trimeric S-protein. PCA was performed using the rotated cryo-EM structures, after converting to the 27-beads model. The PC vectors were calculated upon fitting all the beads. All simulation trajectories were projected onto the PC1 and PC2 vectors and the potential mean forces (PMFs) were calculated for each simulation. The PMF at 310 K in each gREST_SSCR was obtained using all the trajectories using the MBAR method (*Shirts and Chodera, 2008*). The 891 protomers of the cryo-EM structures were also used to compare hinge/twist angles.

## Pocket search and virtual screening

The P2Rank software (*Krivák and Hoksza, 2018*) was used to identify potential druggable pockets in intermediate structures. The cluster centers of 12a, 13a, and 13b as well as $D1_{Sym}$ and $1U_L$ were used for pocket search. All top-ranked pockets were investigated. Pockets at the RBD interface were selected for further analysis as they exist in all three intermediates but vanish in Down and Up, representing potential cryptic pockets. To check the druggability of these pockets, all FDA approved drugs were downloaded from ZINC database (*Irwin and Shoichet, 2005*). This includes 2115 molecules representing 1379 drug candidates. Open Babel was used to convert PDB to PDBQT. AutoDock-Tools-1.5.6 was used to prepare RBD receptor (*Morris et al., 2009*). AutoDock Vina was used to dock all 2115 molecules and perform virtual screening (*Trott and Olson, 2010*).

## Modeling of S-protein structure for MD simulations

The full-length S-protein is formed of three highly glycosylated protomers, with a 1273 residue each. The trimeric structure is divided into three regions, the head region which consists of S1 and part of S2 subunits (residues 1–1140), the heptad repeat 2/the transmembrane domain region (residues 1141–1234), and the cytoplasmic tail region (residues 1235–1273) (*Wrobel et al., 2020*; *Wrapp et al., 2020*). In this study, truncated structures of S-protein including S1 and part of the S2 subunits (residues 28–1135) were used in the simulations. Wherein the starting structure of the Down and Up conformations was based on the cryo-EM structures, PDB:6VXX and the PDB:6VYB, respectively (*Walls et al., 2020*). The 6VXX PDB structure includes multiple missing regions at the NTD (residues 70–79, 144–164, 173–185, and 246–262), the RBD (residues 445–446, 455–461, 469–488, and 502), and the S2 subunit (residues 621–640, 677–688, and 828–853). The 6VYB PDB structure has even more missing regions in the RBD with Up form and the adjacent NTD. Although higher resolution cryo-EM structures were deposited to the PDB later, only these two structures and PDB:6VSB (*Wrapp et al., 2020*) were available when we started this study. Due to the large size and the presence of multiple missing regions, several modeling strategies were used to complete the structures. Wherein, residues 28–292 of NTD were modeled based on the SARS-CoV crystal structure (PDB:5 × 4S at 2.2 Å) (*Yuan et al., 2017*) using Modeller9.19 software (*Sali and Blundell, 1993*). Then part of the modeled region (residues 28–288) was inserted in the cryo-EM structure upon fitting the backbone of residues 263–290. Similarly, the crystal structure of the RBD domain (PDB:6LZG at 2.45 Å) (*Wang et al., 2020*) (residues 336–515) was inserted in the cryo-EM structures upon fitting the Cα atoms of residues 336–400. Both modeled NTD and RBD regions show a good alignment with the resolved regions in the 6VXX and 6VYB cryo-EM structures, see *Figure 1—figure supplement 1*. The VMD program (*Humphrey et al., 1996*) was used to superimpose the modeled regions into the cryo-EM structure. Finally, the missing

regions in the S2 subunit were modeled as loop conformations using the top-ranked structure from Modeller9.19 (*McCallum et al., 2020*). A total of 13 disulfide bonds were included in each protomer including the original 12 disulfide bonds in the cryo-EM structure and one more in the RBD crystal structure. A comparison of our modeled structure and the more recent high-resolution cryo-EM structure (PDB:6ZGE at 2.6 Å) (*Wrobel et al., 2020*) shows a very good agreement, see *Figure 1—figure supplement 1*. Eighteen N-glycans and one O-glycan were added per protomer as suggested in the previous mass spectrometry experiments and a computational model (*Watanabe et al., 2020*; *Woo et al., 2020*). A full list of included glycans is shown in *Figure 1—figure supplement 2*. CHARMM-GUI (*Lee et al., 2016*) was used to make the final model including the addition of glycans, ions (0.15 M NaCl), and water molecules. In total, three S-protein models were built including the Down conformation in the absence of glycan, glycosylated S-protein in Down, and the glycosylated S-protein in Up conformation (*Figure 1—figure supplement 1d*). The total numbers of atoms in each model are 657,411, 654,427, and 654,494, respectively, with the average box lengths of 186.947, 186.452, and 186.475 Å after equilibration, respectively. Finally, the RBD/SD1 monomer models were made by truncating one protomer from the abovementioned Down and Up models, including residues 315–595 (*Figure 2—figure supplement 1b*).

## Further details of gREST_SSCR simulations

We recently proposed an enhance sampling method, the generalized replica exchange with solute tempering of selected surface charged residue (gREST_SSCR) (*Dokainish and Sugita, 2021*) to enhance large domain motions in multi-domain proteins. In this method the Coulomb and Lennard Jones parameters of surface charged residues at the domain interfaces are selected as a solute region in gREST (*Kamiya and Sugita, 2018*). In this study, to enhance conformational dynamics of S-protein, we performed gREST_SSCR simulations, wherein charged residues at the interfaces between two RBD domains, between RBD and NTD, and between RBD and S2, were selected as the solute region (*Figure 1—figure supplement 3a*). In total, 16 residues in each protomer, consisting of 8 positive and 8 negative charged residues (870 atoms), were selected as solute in gREST: K113, K378, K386, R408, K417, K462, R466, R983, E132, E169, D198, D405, E406, D420, D428, and E471. We performed two sets of 100 ns cMD simulations from Down and Up cryo-EM conformations in the presence and absence of positional restraints on backbone atoms, respectively, and carried out a preliminary gREST simulation where all charged residues in one of RBDs were included in the solute region. H-bonding and contact analysis were performed to identify the abovementioned residues in the solute region. Some pairs (e.g. D428_R983) represented the native contacts observed in the restrained cMD simulations, Down$_{Asym}$ in the cMD simulation in the absence of restraints included the salt-bridge pair, K462_D198, and newly formed salt-bridge pairs (e.g. K378_D405) were found in the preliminary gREST simulation. In summary, we carefully selected essential charged residues from the cryo-EM structures, preliminary cMD and gREST simulations to accelerate large conformational changes of RBDs in S-protein. All simulations were performed using 16 replicas covering a solute temperature parameter range from 310.00 to 545.00 K while maintaining solvent temperature at 310.15 K in NVT ensemble. We carried out three gREST_SSCR simulations: two from Down in the presence (500 ns) and absence of glycans (150 ns), and one from Up (300 ns). The total simulation times correspond to 8, 4.8, and 2.4 μs in gREST_Down, gREST_Up, and gREST_Down w/o glycan, respectively.

All simulations were performed using the new version of GENESIS MD software that was optimized on Fugaku (*Kobayashi et al., 2017*; *Jung et al., 2021*). The overall performance of gREST_SSCR simulations using 16 replicas is 52 ns/day using 2048 nodes on Fugaku. CHARMM 36m force field was used for protein (C36m), carbohydrate, and ions, while CHARMM TIP3P was used as a water model (*Huang et al., 2017*; *Guvench et al., 2011*). gREST_SSCR simulations were performed after a series of equilibration steps. First modeled systems were minimized for 10,000 steps, while applying positional restraint on the backbone atoms. Second, using leap-frog integrator and the Langevin thermostat, we heated the simulation systems to 310.15 K in a stepwise manner for 100 ps. Third, a series of equilibration steps were performed: (1) MD simulations in the NVT ensemble using the velocity Verlet integrator with stochastic velocity rescaling thermostat (*Bussi et al., 2007*). (2) Those in the NPT ensemble with stochastic velocity rescaling thermostat and MTK barostat (*Bussi et al., 2007*; *Bussi et al., 2009*; *Jung et al., 2018*) (note that all previous steps also included a weak restraints on side chain and glycan dihedral angles). (3) After removing all restraints, another MD simulation in the NPT ensemble

was performed as equilibration using the same protocol. (4) MD simulation in the NVT ensemble was followed as the second equilibration using the same thermostat and the multiple time step integrator with a fast motion time step of 2.5 fs, and slow motion every 5 fs (*Tuckerman et al., 1992*). (5) Prior to production run, a 2 ns equilibration was performed for 16 replicas. Production runs were then performed for 150, 500, and 300 ns per replica in gREST_Down w/o glycan, gREST_Down, and gREST_Up simulations, respectively. At every 20 ps, replica exchanges were attempted, and trajectories were saved. Electrostatic interactions were computed by smooth particle mesh Ewald (*Essmann et al., 1995*) method with 128 × 128 × 128 grids and the sixth-order B-spline function. Temperature is evaluated using the group-based approach with an optimal temperature evaluation, and thermostat is applied at every 10 steps (*Jung and Sugita, 2020*). Classical MD simulation of RBD/SD1 monomer structures was performed for 300 ns. Two independent simulations were performed starting from Up and one from Down. In all simulations, water molecules were constrained with SETTLE, while bonds involving hydrogens were constrained with SHAKE/RATTLE algorithm (*Miyamoto and Kollman, 1992*; *Ryckaert et al., 1977*; *Andersen, 1983*).

## Simulation trajectory analysis

To characterize the RBD motions, two main criteria are considered: the Cα atoms RMSD of RBD upon fitting the S2 Cα atoms of cryo-EM structure (residues 689–827 and 854–1134) and the RBD hinge and twist angles. Hinge and twist angles represent relative domain motions of RBD, wherein the hinge angle describes the Down/Up transition while the twist angle describes RBD side motion. The hinge angle is defined with three points, the center of mass of the Cα atoms in the SD1 core (residues 324–329, 531–590), the top residues of SD1 (residues 328, 329, 530, 531, 543, and 544), and the center of mass in the RBD core (residues 335–466 and 491–526). To define the twist angle, one more point was added at the lower part of RBD (residues 335, 336, 361, 362, 524, and 525). To examine the intra-domain stability, we computed the Cα atom RMSD of RBD (residues 333–528) and NTD (residues 28–306). The Cα RMSD of RBD and NTD upon fitting to their own structures at 310 K in gREST_Down reveal about 1.4 and 2.2 Å, respectively (*Figure 1—figure supplement 5a and b*). They are comparable to those in the cMD (*Mori et al., 2021*) (1.4 and 1.6 Å for RBD and NTD, respectively). Slightly larger RMSD values of NTD in gREST_SSCR are attributed to the loop regions abundant in NTD, as indicated in root mean square fluctuations (RMSF) (*Figure 1—figure supplement 5c and d*). The Cα RMSD of the simulations' representative structures (Down$_{Sym}$, 1U, 1U$_O$, and 2U$_L$) were calculated with respect to cryo-EM structures using all three chains including 2766, 2884, 2450, and 2719 Cα atoms, respectively. Since 1U$_O$ is compared to three-Up cryo-EM structure, two RBDs with a Down form were excluded in the RMSD calculation.

The k-means algorithm in GENESIS software package was used to classify the conformations of S-protein in MD simulations at 310 K. Hereafter, all the analysis is carried out to obtain the canonical ensembles at 310 K. The number of clusters in k-means clustering was set to eight in all cases. The cluster analysis was performed with the same fittings used in the RMSD analysis of RBD. Only the Cα atoms included in the original cryo-EM structures (PDB:6VXX and PDB:6VYB) were included to avoid flexible regions in our analysis. Furthermore, the distributions of hinge and twist angles for all the eight clusters were calculated for each protomer (in total, six angles) and subsequently the number of clusters was increased until the hinge/twist distribution showed the minimal overlaps (*Figure 4—figure supplements 1–3*). In this procedure, 12, 13, and 13 clusters were obtained in gREST_Down w/o glycan, gREST_Down, and gREST_Up simulations, respectively (*Supplementary file 1C*).

Due to the homo-trimeric nature of S-protein, protomers are indistinguishable in the structure. gREST_SSCR enhanced motions of RBD regions so that we don't know which protomer reveals large-scale conformational motions in any replicas. For instance, *Figure 1—figure supplement 6b* shows that RBD$_A$ undergoes large transition in replica 1, while RBD$_C$ shows large motion in replica 16. To clarify the discussion in this paper, we applied a rotational scheme that makes RBD$_A$ undergo the largest conformational transitions in the following ways: (1) We identify all replicas that show significant RBD motions with a hinge angle >130° in RBD$_B$ or RBD$_C$. (2) We rotate the conformations of those selected replicas where RBD$_B$ or RBD$_C$ becomes RBD$_A$ while rotating the rest of the molecule including glycans (*Figure 1—figure supplement 6a*). (3) We confirm the rotation scheme by comparing hinge/twist angle free-energy maps before and after rotation (*Figure 1—figure supplement 6c and d*, 7a, and 7b). We also compared PCA before and after rotations. (4) In cases of two

RBDs showing large hinge angles in the same replica, the protomer with the highest RBD hinge angle becomes Chain$_A$.

HB and contact analysis were also performed for major clusters, wherein a 75% and 50% probability threshold were used for the heavy atoms' contacts and HB residue pairs selection in *Figure 4—figure supplements 7 and 8*, respectively. The correspondence analysis to the previous smFRET experiment (*Lu et al., 2020*) was performed upon calculating the COM distance from residues 425–431 to residues 554–561 using the Cα atoms (*Figure 4—figure supplement 4a*). Experimental statistical ratio (*Lu et al., 2020*) of 77% and 23% was used to combine gREST_Down and gREST_Up simulation results, respectively, as shown in *Figure 4—figure supplement 4d*. The VMD and PyMOL programs were used for trajectory and structure visualization (*Humphrey et al., 1996*; *Schrodinger LLC, 2021*).

SASA values were calculated using the measure SASA function in VMD (*Humphrey et al., 1996*). The restrict option, which considers only solvent accessible points near the user specified region, was used for per-domain and per-residue SASA calculations. A range of probe radius, including 1.4 Å (a sphere of water) and 7.2 Å (approximating the hypervariable loops of antip-gp120 antibodies) (*Grant et al., 2020*), was used for SASA calculations. SASA values were calculated for different RBD conformations, Down, 1U, 1U$_O$, and 2U$_L$. Down represent the sum of Down$_{Sym}$ and Down$_{Asym}$, the rest is defined in *Supplementary file 1*. For each conformation, 30 snapshots close to the cluster center were extracted and used for analysis. The calculated SASA values were mapped on the structure using PyMOL software (*Schrodinger LLC, 2021*). For comparison, we confirmed that our SASA calculations give the results in consistent with the previous work by Amaro and co-workers (*Figure 3—figure supplement 1*; *Casalino et al., 2020*).

# Acknowledgements

This research used computational resources of the supercomputer Fugaku (the evaluation environment in the trial phase) provided by the RIKEN Center for Computational Science. The results obtained on the evaluation environment in the trial phase do not guarantee the performance, power, and other attributes of the supercomputer Fugaku at the start of its public use operation. The computer resources of Oakforest-PACS were also provided through HPCI System Research project (Project ID: hp200135, hp200153, hp200028, hp210107, hp210177).

# Additional information

## Competing interests

## Funding

| Funder | Grant reference number | Author |
|---|---|---|
| Ministry of Education, Culture, Sports, Science and Technology | FLAGSHIP 2020 project | Yuji Sugita |
| Ministry of Education, Culture, Sports, Science and Technology | Priority Issue on Post-K computer | Yuji Sugita |
| Ministry of Education, Culture, Sports, Science and Technology | Program for Promoting Researches on the Supercomputer Fugaku | Yuji Sugita |
| Ministry of Education, Culture, Sports, Science and Technology | JPMXP1020200101 | Yuji Sugita |
| Ministry of Education, Culture, Sports, Science and Technology | JPMXP1020200201 | Yuji Sugita |

| Funder | Grant reference number | Author |
|---|---|---|
| Ministry of Education, Culture, Sports, Science and Technology | 19H05645 | Yuji Sugita |
| Ministry of Education, Culture, Sports, Science and Technology | 21H05249 | Yuji Sugita |
| Ministry of Education, Culture, Sports, Science and Technology | 20K15737 | Hisham M Dokainish |
| Ministry of Education, Culture, Sports, Science and Technology | 19K12229 | Suyong Re |
| Ministry of Education, Culture, Sports, Science and Technology | 21H05157 | Takaharu Mori |
| Ministry of Education, Culture, Sports, Science and Technology | 19K06532 | Takaharu Mori |
| RIKEN | Dynamic Structural Biology/Glycolipidologue Initiative/Biology of Intracellular Environments | Yuji Sugita |
| HPCI System Research project | hp200135 | Takaharu Mori |
| HPCI System Research project | hp200153 | Takaharu Mori |
| HPCI System Research project | hp200028 | Takaharu Mori |
| HPCI System Research project | hp210107 | Takaharu Mori |
| HPCI System Research project | hp210177 | Takaharu Mori |

The funders had no role in study design, data collection and interpretation, or the decision to submit the work for publication.

## Author contributions

Hisham M Dokainish, Conceptualization, Data curation, Formal analysis, Investigation, Methodology, Project administration, Validation, Visualization, Writing - original draft, Writing - review and editing; Suyong Re, Formal analysis, Methodology, Visualization, Writing - original draft, Writing - review and editing; Takaharu Mori, Formal analysis, Methodology, Software, Validation, Visualization; Chigusa Kobayashi, Formal analysis, Methodology, Software, Visualization; Jaewoon Jung, Methodology, Software; Yuji Sugita, Conceptualization, Funding acquisition, Methodology, Project administration, Resources, Supervision, Writing - review and editing

## Author ORCIDs

Hisham M Dokainish ⬤ http://orcid.org/0000-0002-4387-4790
Suyong Re ⬤ http://orcid.org/0000-0002-3752-6554
Takaharu Mori ⬤ http://orcid.org/0000-0002-8717-2926
Chigusa Kobayashi ⬤ http://orcid.org/0000-0002-5603-4619
Jaewoon Jung ⬤ http://orcid.org/0000-0002-2285-4432
Yuji Sugita ⬤ http://orcid.org/0000-0001-9738-9216

## Decision letter and Author response

Decision letter https://doi.org/10.7554/eLife.75720.sa1
Author response https://doi.org/10.7554/eLife.75720.sa2

## Additional files

### Supplementary files

• Supplementary file 1. List of PDBs, clusters and lignads. (A) Cryo-electron microscopy (cryo-EM) structures used in the principal component analysis (PCA). (B) Definition of protomer coarse-grained particles representing rigid domains for PCA. (C) List of clusters for gREST_Down, gREST_Up, and gREST_Down w/o glycan simulations. (D) The receptor binding domain (RBD) interface cryptic pockets predicted by P2Rank. (E) List of the top-ranked molecules from the virtual screening of 2115 FDA approved drugs to RBD interface in I2a, I3a, and I3b intermediate structures. (F) Nilotinib binding energy to I2a, I3a, and I3b intermediates.

• Transparent reporting form

### Data availability

The trajectories were computed with GENESIS 2.0 beta, open source program https://www.r-ccs.riken.jp/labs/cbrt/ and analyzed using GENESIS 1.6.0 analysis tools https://www.r-ccs.riken.jp/labs/cbrt/download/genesis-version-1-6/. Simulation data were deposited at https://covid.molssi.org/. Data of gREST simulations from Down and data of gREST_Up simulations, including model and simulation structures, are available.

The following datasets were generated:

| Author(s) | Year | Dataset title | Dataset URL | Database and Identifier |
|---|---|---|---|---|
| Suyong R, Hisham MD, Takaharu M, Chigusa K, Jaewoon J, Yuji S | 2021 | gREST_SSCR Simulation of Trimeric SARS-CoV-2 Spike Protein Starting From Down Conformation | https://doi.org/10.34974/wtbx-0r84 | MolSSI, 10.34974/wtbx-0r84 |
| Suyong R, Hisham MD, Takaharu M | 2021 | gREST_SSCR Simulation of Trimeric SARS-CoV-2 Spike Protein Starting From 1Up Conformation | https://doi.org/10.34974/xn67-xk26 | MolSSI, 10.34974/xn67-xk26 |

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
