## [Editor Report]

Using extensive molecular dynamics simulations with a novel enhanced sampling technique, the authors are able to characterize the structural flexibility of the SARS-CoV2 spike protein and identify new conformational states. These insights will be valuable to the design of novel strategies that modulate the interactions of the spike protein during the infection process.

---

## [Decision Letter]

**Decision letter after peer review:**

Thank you for submitting your article "The Inherent Flexibility of Receptor Binding Domains in SARS-CoV-2 Spike Protein" for consideration by *eLife*. Your article has been reviewed by 3 peer reviewers, one of whom is a member of our Board of Reviewing Editors, and the evaluation has been overseen by a Reviewing Editor and Volker Dötsch as the Senior Editor. The reviewers have opted to remain anonymous.

Essential revisions:

1) More thorough discussion of the conformational transition pathways, including key intermediates and their structural stability.

2) More quantitative analysis of the key structural states identified in the study, evaluation of their potential accuracy, especially the intermediate states for which limited high-resolution information is available from experiments so far.

3) Considering the publication of several recent computational analyses of the same system, it would be important to further highlight the unique insights from the current analysis.

*Reviewer #1:*

Understanding the conformational dynamics of the spike protein of SARS-CoV-2 can provide valuable guidance to the design of new antivirial drugs and vaccines. Previous computational studies employed either bias or enhanced sampling along specific structural variables, leading to incomplete understanding in the structural flexibility and conformational transition mechanisms. In this study, the authors take advantage of a newly developed enhanced sampling method (gREST_SSCR), which does not specify any biasing coordinates, to better characterize the structural ensemble of the spike protein. The results therefore complement previous simulations studies and enhance our understanding in the conformational flexibility and transition mechanism of the spike protein. For example, the simulations identified intermediate states that were observed in smFRET experiments but not in higher-resolution cryo-EM studies. The sampled conformational ensembles also revealed cryptic binding sites that potentially bind to drug molecules and binding interfaces of antibodies. The results will be instructive to the design of new approaches for battling this important pandemic.

While I find the simulations impressive and analyses rather comprehensive, I hope the authors would consider the following questions.

1. For comparing with the cryo-EM structures in Figure 2 – can the comparison be made more quantitative?

2. On the criterion for selecting the "solute" residues – it is understandable that charged residues at domain interfaces might be good choices since interactions involving them are likely to be reorganized during large-scale conformational transitions. Nevertheless, it will be useful to elaborate how the specific set of residues are chosen. For example, are they involved in different salt-bridge interactions in different conformational states identified in the cryo-EM structures?

3. Considering recent publications that also highlighted conformational flexibility of the spike protein, roles of glycan in modulating conformational flexibility/transition and binding interface with other proteins, and cryptic binding sites of small molecules, it would be important to clearly highlight how observations from the current work differ from these complementary computational efforts.

*Reviewer #2:*

In the manuscript by Sugita and colleagues, the authors describe the results of extensive simulations of the spike protein of SARS-CoV-2. They used generalized replica exchange with solute tempering of selected surface charged residues (gREST_SSCR) to study the conformational changes of the SARS-CoV-2 spike (S) protein, specifically the up and down motion of the receptor binding domains (RBDs). They observed the Down, one-Up, one-Open and two-Up-like structures in their simulations, and compared them with the cryo-EM structures. The solvent accessible surface area (SASA) of the RBD was measured at each of the above states to study the effect of different conformations for ACE2 and antibodies binding. They further used k-means clustering and re-clustering to obtain five main conformation ensembles, providing structural insight to a previous smFRET experiment and, more importantly, the transition pathway from the Down to one-Up state. The pathway was characterized by the analysis of contacts and hydrogen bonds. At the end, the authors applied P2Rank to search for the formation of druggable pockets in the intermediate structures.

Overall I found the simulation and analysis methods used in this study very impressive and the findings are important. However, I found the Results section of the manuscript to be rather confusing, making it difficult to capture the major findings of the studies until I read the Discussion section.

1) The definitions of the many different spike conformations throughout the manuscript were the most confusing part for me. The first four structures (D_sym, 1U, 1U_o, and 2U_L) are the most well defined, with their hingeA and hingeB value provided and a clear figure (Figure 2) showing their 3D structures as well as where they located in the PC space and Hinge angle space. However, things start getting murkier when it comes to the 5 ensembles (Down_sym, Down_like, Int2, Int3 and 1Up) found by k-means clustering. It was not discussed in the main text whether Down_sym is the same state as the D_sym state defined before, or how Down_like is different. It is also not clear what Int2 and Int3 states look like. Only a careful reader would find there are actually more than 5 clusters and their cluster centers can be interpreted from Figure 4—figure supplement 1, and they are grouped into 5 major conformations under the scheme from Figure 4—figure supplement 4e, and their respective distributions on the Hinge angle space was plotted at Figure 4—figure supplement 5. It is not helping that the authors go on referencing these minor clusters in the main text (one-Up-like, 1U_L and 1U_a in the main text, IUa, 2Ua_L, 2Ub_L in Figure 5) without providing their definitions. I believe such information is crucial for the reader to understand the second half of the manuscript, and I suggest the authors reorganize the figures to make the information clear and easy to access.

2) The second point I want to make goes hand-in-hand with the first point. I find the transition paths and the intermediate states to be the most interesting and novel findings of this work, yet the discussions on these findings are very brief in the manuscript. How does the hinge and twist angles change when you go from one state to another? What are the roles of the intermediate states (Int2 and Int3) in this opening of the RBD? I noticed in Figure 4—figure supplement 5(b) there is no arrow pointing from state I3b to 1U_L. Is I3b a trap state? How did the authors come to this conclusion? How does I3a differ from I3b? What about the one-Open state that the authors suggest to be important for binding? The discussion on the 1Up-to-2Up transition is equally brief. I believe these questions deserve a much longer discussion than a brief page-and-a-half skim over.

*Reviewer #3:*

This article focuses on the varied conformations that the SARS-CoV-2 spike protein takes and how this relates to the function of this protein and potential for drug targets. Extensive molecular simulations at the atomistic level were performed to provide details of the structural transitions of the spike protein and its protective sugar shield. Overall, this manuscript is a strong study on the spike structural states and transitions using enhanced molecular dynamics. This work clearly lays out details of transitions near the down and up states and how glycans can stabilize or alter conformational states and transitions. This work adds important details to the growing body of literature on the spike protein.

The main weakness in this work is the focus on intermediate structures in drug binding. Although this is an interesting idea, the details of the stability of the intermediate states are not described. Stability is key to stable for drug binding to deactivate the spike protein. It is unclear without more detail if these intermediate states are likely to be stable for drug binding and deactivation.

Author Recommendations (details):

1. Lack of details for intermediate states: I don't see analysis showing intermediate stability of I2a and I3a. How stable are these and where are they on the reduced free energy surface in Figure 1? Short-lived intermediates are not good targets for drugs. Figure 4 provides some details in the structure but no info on the free energy surface near the intermediates is provided and how this relates to stability.

2. Accuracy of intermediate state: How do you know these are accurate intermediate states? Any experimental comparison that would suggest these exist?

3. Is spike a good drug target?: The spike protein is certainly an important target for antibodies and vaccines. However, it may not be the best target toward drugs and a current FDA approved drug focuses on another part of the virus action. Some discussion on why the spike protein is believed to be a good target should be included in the manuscript.

4. Supporting figure format: I find it difficult to read and follow the format for the supporting figures that are tied to a main figure and not in a single pdf document. I would hope that *eLife* allows a single supporting document with all the figures numbered to easily review like all other journals.

5. Add omicron: With the current state of SARS-CoV-2, the list of mutants/variants in the intro should now include omicron.

---

## [Author Response]

Essential revisions:1) More thorough discussion of the conformational transition pathways, including key intermediates and their structural stability.

We added discussion about the conformational transition pathways (down-to-1up as well as 1upto-2up transitions) in the revised manuscript. For this purpose, we used conformational distributions projected on the Hinge-Hinge or Hinge-Twist angles. They are specifically addressed in our replies to the 2^nd^ and 3^rd^ comments from reviewer #2.

2) More quantitative analysis of the key structural states identified in the study, evaluation of their potential accuracy, especially the intermediate states for which limited high-resolution information is available from experiments so far.

We added quantitative analysis and compared the intermediate structures with available cryo-EM structures carefully. Due to the transient nature of intermediate structures, we carefully discuss what we have observed in intermediate structures (domain motions and molecular interactions). They are addressed in our replies to the 1^st^ comment from reviewer #1 and the 2^nd^ comment from reviewer #3.

3) Considering the publication of several recent computational analyses of the same system, it would be important to further highlight the unique insights from the current analysis.

In the original manuscript, we have cited most of the important computational papers about the spike protein dynamics. But, we highlighted the uniqueness of our work at the end of Discussion of the revised manuscript. This is specially discussed in our reply to the 3^rd^ comment from reviewer #1.

Reviewer #1:Understanding the conformational dynamics of the spike protein of SARS-CoV-2 can provide valuable guidance to the design of new antivirial drugs and vaccines. Previous computational studies employed either bias or enhanced sampling along specific structural variables, leading to incomplete understanding in the structural flexibility and conformational transition mechanisms. In this study, the authors take advantage of a newly developed enhanced sampling method (gREST_SSCR), which does not specify any biasing coordinates, to better characterize the structural ensemble of the spike protein. The results therefore complement previous simulations studies and enhance our understanding in the conformational flexibility and transition mechanism of the spike protein. For example, the simulations identified intermediate states that were observed in smFRET experiments but not in higher-resolution cryo-EM studies. The sampled conformational ensembles also revealed cryptic binding sites that potentially bind to drug molecules and binding interfaces of antibodies. The results will be instructive to the design of new approaches for battling this important pandemic.

We are also happy to hear the positive comments of reviewer #1 about the public importance of our paper.

While I find the simulations impressive and analyses rather comprehensive, I hope the authors would consider the following questions.1. For comparing with the cryo-EM structures in Figure 2 – can the comparison be made more quantitative?

To compare the simulated structures with the cryo-EM structures more quantitatively, the Ca RMSD between them are computed. On page 6, lines 143-150 in the revised manuscript, we modified:

“They are superimposed to the cryo-EM structures having corresponding S-protein conformations in Figure 2d. Down_Sym_ and 1U from our simulations are well aligned to the high-resolution cryoEM structures of Down (PDB:6ZGE^16^ (RMSD: 3.7Å)) and one-Up (PDB:6XKL^17^ (RMSD: 3.6Å)), respectively. Intriguingly, RBD_A_ in 1U_O_ is aligned to one of the RBDs in one of the three-Up structures (PDB:7DCC^45^ (RMSD: 5.4 Å)). Although 2U_L_ from our simulations remains some interactions between RBD_A_ and RBD_B_, which is completely lost in two-Up cryo-EM structures, 2U_L_ is aligned with those in one of the two-Up structures (PDB:6X2B^43^ (RMSD: 5.8 Å)). The overall comparison of the simulations’ structures to cryo-EM points out the quality of the predicted structures.”

On Page 21 and lines 522-525 in the revised manuscript, the following sentences of the analysis methods were added:

“The Ca RMSD of the simulations’ representative structures (Down_Sym_, 1U, 1U_O_ and 2U_L_) were calculated with respect to cryo-EM structures using all three chains including 2766, 2884, 2450 and 2719 Ca atoms, respectively. Since 1U_O_ is compared to three-Up cryo-EM structure, two RBDs with a down form were excluded in the RMSD calculation.”

2. On the criterion for selecting the "solute" residues – it is understandable that charged residues at domain interfaces might be good choices since interactions involving them are likely to be reorganized during large-scale conformational transitions. Nevertheless, it will be useful to elaborate how the specific set of residues are chosen. For example, are they involved in different salt-bridge interactions in different conformational states identified in the cryo-EM structures?

The criterion for selecting charged residues “solute” were based on four preliminary cMD simulations of the Down and Up conformations, in the presence and absence of backbone restraints for 100 ns each. Contact and H-bonding analysis were performed to select salt bridge pairs from cMD simulations. This list was further refined based on the results of a preliminary gREST simulation where all charged residues in RBD_B_ were included in the solute region. Finally, the solute list was based on comparing native contact in cryo-EM, cMD and gREST simulation. On page 19, line 474-482 in the revised manuscript, the following sentences were added:

“We performed two sets of 100ns cMD simulations from Down and Up cryo-EM conformations in the presence and absence of positional restraints on backbone atoms, respectively, and carried out a preliminary gREST simulation where all charged residues in one of RBDs was included in the solute region. H-bonding and contact analysis were performed to identify the above-mentioned residues in the solute region. Some pairs (e.g. D428_R983) represented the native contacts observed in the restrained cMD simulations, Down_Asym_ in the cMD simulation in the absence of restraints included the salt-bridge pair, K462_D198, and newly formed salt-bridge pairs (e.g. K378_D405) were found in the preliminary gREST simulation. In summary, we carefully selected essential charged residues from the cryo-EM structures, preliminary cMD and gREST simulations to accelerate large conformational changes of RBDs in spike protein”

3. Considering recent publications that also highlighted conformational flexibility of the spike protein, roles of glycan in modulating conformational flexibility/transition and binding interface with other proteins, and cryptic binding sites of small molecules, it would be important to clearly highlight how observations from the current work differ from these complementary computational efforts.

We consider that our work has two unique features based on our simulation and analysis methods. gREST simulations could avoid the use of reaction coordinates or collective variables, which might limit the enhanced conformational sampling space to the limited one. Instead, gREST simulations where only the atomic interactions related to the selected solute region is scaled, include less biases in the simulation trajectories and allow us to observe unexpected conformational changes, such as two-Up RBD structures, Down_Asym_, etc. Due to their transient nature, these structures are difficult to be observed by conventional experiments and simulations.

The transient structures include novel interactions in a spike protein, which are not observed in cryo-EM structures. To clarify the atomic interactions in each intermediate or transient structures is important, while it was not analyzed in the previous computational works. Instead, we quantitatively analyzed hydrogen bonds and hydrophobic contacts in the residues and glycans of intermediate structures observed in gREST, which allow us to define conformational transition pathways and understand molecular mechanisms for the transition in the atomic detail. In the end of Discussion, on page 15, lines 377-392, we added the following sentences:

“Finally, we highlight the key differences between our simulations and other computational works on spike protein. Current work includes at least two unique features based on the simulation and analysis methods. First, our conformational sampling method, gREST, could avoid the use of predefined reaction coordinates and/or bias potentials along the coordinates. Therefore, the gREST simulation is able to generate unexpected structures or structural changes, which are free from the starting cryo-EM structures and the reaction coordinates in biased MD simulations. Two-Up like structures, Down_Asym_, or 1U_O_ were not expected even by us before the extensive gREST simulations and analysis. Since these unexpected structures involve novel molecular interactions, it is necessary to detect which interactions can stabilize the intermediate structures and to examine how the interactions are changed along the transition pathways. Therefore, we analyzed the hydrogen bonds and contacts in each intermediate, systematically, and quantitatively (the second key feature). The changes of the atomistic interactions involving side chains and glycans could characterize the conformational transition pathways in atomic detail without using simplified reaction coordinates. To understand the effect of mutations in the conformational changes toward the active Up form, the difference of intrinsic flexibility between wild type and mutant spike protein is one of the most important features to be understood. The current simulation and analysis methods are applicable to new variants for giving atomistic pictures of conformational flexibility and transitions of spike proteins, which are both important for understanding the essential characters of new spike protein variants.”

Reviewer #2 (Recommendations for the authors):In the manuscript by Sugita and colleagues, the authors describe the results of extensive simulations of the spike protein of SARS-CoV-2. They used generalized replica exchange with solute tempering of selected surface charged residues (gREST_SSCR) to study the conformational changes of the SARS-CoV-2 spike (S) protein, specifically the up and down motion of the receptor binding domains (RBDs). They observed the Down, one-Up, one-Open and two-Up-like structures in their simulations, and compared them with the cryo-EM structures. The solvent accessible surface area (SASA) of the RBD was measured at each of the above states to study the effect of different conformations for ACE2 and antibodies binding. They further used k-means clustering and re-clustering to obtain five main conformation ensembles, providing structural insight to a previous smFRET experiment and, more importantly, the transition pathway from the Down to one-Up state. The pathway was characterized by the analysis of contacts and hydrogen bonds. At the end, the authors applied P2Rank to search for the formation of druggable pockets in the intermediate structures.Overall I found the simulation and analysis methods used in this study very impressive and the findings are important. However, I found the Results section of the manuscript to be rather confusing, making it difficult to capture the major findings of the studies until I read the Discussion section.

Thank you for giving us positive comments about our work. In the revised manuscript, we tried to explain our major findings as clear as possible.

1) The definitions of the many different spike conformations throughout the manuscript were the most confusing part for me. The first four structures (D_sym, 1U, 1U_o, and 2U_L) are the most well defined, with their hingeA and hingeB value provided and a clear figure (Figure 2) showing their 3D structures as well as where they located in the PC space and Hinge angle space. However, things start getting murkier when it comes to the 5 ensembles (Down_sym, Down_like, Int2, Int3 and 1Up) found by k-means clustering. It was not discussed in the main text whether Down_sym is the same state as the D_sym state defined before, or how Down_like is different. It is also not clear what Int2 and Int3 states look like. Only a careful reader would find there are actually more than 5 clusters and their cluster centers can be interpreted from Figure 4—figure supplement 1, and they are grouped into 5 major conformations under the scheme from Figure 4—figure supplement 4e, and their respective distributions on the Hinge angle space was plotted at Figure 4—figure supplement 5. It is not helping that the authors go on referencing these minor clusters in the main text (one-Up-like, 1U_L and 1U_a in the main text, IUa, 2Ua_L, 2Ub_L in Figure 5) without providing their definitions. I believe such information is crucial for the reader to understand the second half of the manuscript, and I suggest the authors reorganize the figures to make the information clear and easy to access.

We apologize for making the reviewer being confused because of the lack of cluster definitions. In the revised manuscript, we added the cluster definitions and linked the macro-clusters with micro ones in Table 3 of the supplementary file 1C. Since D_Sym_ is the same conformation as Down_Sym_, we changed all the D_Sym_ to Down_Sym_ in the revised manuscript. In the revised manuscript on page 8, lines 194-201, the following description was added:

“The 13 micro-clusters were identified from gREST_Down (Figure 4—figure supplement 1) including two clusters representing the symmetric Down (Down_sym: D1_Sym_ and D2_Sym_), two clusters of asymmetric Down with differences in RBDs twist angle distribution (Down_Asym_: D1_asym_ and D2_asym_), three clusters with a slight increase in the RBD hinge angle around 120°, (Int1: I1a, I1b and I1c), three clusters with one RBD hinge angle around 130° (Int2: I2a, I2b and I2c), two clusters with hinge angle around 140° (Int3: I3a and I3b), and one cluster with an Up-like conformation (1U_L_). Similarly, from gREST_Up, the 13 clusters were identified including ten clusters representing one-Up conformation (1U: 1Ua-1Uj), two clusters with two-Up like conformations (2U_L_: 2Ua_L_ and 2Ub_L_) and one-Up open cluster (1U_O_).”

One page 9, lines 203-205 in the revised manuscript, we modified the sentence as follows:

By combining the distance distributions from gREST_Down and gREST_Up, we obtain five main conformational ensembles (Figure 4—figure supplement 4): Down symmetric (Down_Sym_), Down like (Down_Like_: Down_Asym_ and Int1), two intermediates (Int2 and Int3), and Up (1Up).

2) The second point I want to make goes hand-in-hand with the first point. I find the transition paths and the intermediate states to be the most interesting and novel findings of this work, yet the discussions on these findings are very brief in the manuscript. How does the hinge and twist angles change when you go from one state to another? What are the roles of the intermediate states (Int2 and Int3) in this opening of the RBD? I noticed in Figure 4—figure supplement 5(b) there is no arrow pointing from state I3b to 1U_L. Is I3b a trap state? How did the authors come to this conclusion? How does I3a differ from I3b? What about the one-Open state that the authors suggest to be important for binding? The discussion on the 1Up-to-2Up transition is equally brief. I believe these questions deserve a much longer discussion than a brief page-and-a-half skim over.

Thank you for pointing out the important issues to clarify our key findings in the transition paths and intermediate states in the Down-to-1Up transition. To answer the questions raised by the reviewer #2, we added sentences in the section (On page 9, lines 215-231) in the revised manuscript:

“We next focus on molecular mechanisms underlying the Down-to-1Up transitions in terms of the correlated motions of the hinge and twist angles in the three RBDs (Figure 4—figure supplement 5). Note that such correlated motions are hardly obtained using targeted MD simulations (Figure 4—figure supplement 6) or the MD simulations enhanced with pre-defined reaction coordinates^40, 48^. Figure 4—figure supplement 5 indicates that the transition pathway from Down-to-1Up conformation occurs via four main states including two intermediates. First, the transition is initiated from flexible Down structures (Down_Sym_ and Down_Like_), where three RBDs show high flexibilities reflected by a wide range of twist angle between 10° and 90° while RBD_A_ maintains a hinge angle < 125°. The increase in RBD_A_ hinge angle to 130° in I2a is accompanied by the reduction in other RBDs flexibility. It is reflected by smaller changes in twist and hinge angles for the chains B and C. Later, two main pathways are identified through I3a or I3b (hinge 140°), however comparisons of all five free energy maps as well as the projection on PC1-PC2 free-energy surface suggest that 13b is an off-target intermediate. In I3a, both hinge and twist angles are increased in RBD_A_, accompanied by a slight reduction in RBD_C_ hinge angle. Both I2a and I3a represent stable basins along the free-energy landscapes along the PC1-PC2 and Hinge-Twist angles, emphasizing their role in mediating the transition (Figure 4—figure supplement 5 and 11). Lastly, 1U_L_ conformation is formed with hinge angle > 150° with a concomitant increase in its twist angle. In general, Hinge_A_ and Twist_A_ are highly correlated (cc: 0.94) with each other (Figure 4—figure supplement 5c), while Hinge_A_ and Hinge_B_ show no correlations (cc: 0.02). The correlations between Hinge_Å_ and Hinge_C_/Twist_C_ exist to some extent (cc: -0.68 and 0.30).”

A new supplementary figure (Figure 4—figure supplement 11) was added.

3. The 1Up-2Up transition pathway was also further discussed

We discussed the 1Up-2Up transition pathways more carefully on page 10 lines 242-254 in the revised manuscript:

“The formation of two Up-like conformation is characterized by the increase of Hinge/Twist angle in RBD_B,_ as shown in Figure 1—figure supplement 7c. FEL on Hinge_A_-Twist_A_ angles shows a similar transition as observed in RBD_A_ in gREST_Down (Figure 1—figure supplement 7c and 6d). The relative RBD motions during the 1Up-to-2Up transition is characterized using the hinge angles (Figure 5—figure supplement 1a). The increase of hinge angle in RBD_B_ is accompanied with a slight decrease of hinge angle in RBD_A_ in 2Ua_L_ and subsequently in 2Ub_L_. Note that both clusters might represent intermediate states toward the full 2Up structure, in which the increase of RBD_A_ hinge angle is regained. The latter decrease likely relates with the increases of the contacts and HBs between RBD_A_ and RBD_C_, for instance, K378_A_-E484_C_, Y369_A_-N487_C_ and D427_A_-Y505_C_, from 1Ua, the top populated cluster in 1U, to 2U_L_ structures (Figure 5 and Figure 4—figure supplement 7 and 8). The numbers of contacts and HBs in two-Up-like structures are much less than those in 1Ua, suggesting that a drastic reduction of the inter-domain interactions between RBD_A_ and RBD_B_ is required to form a full two-Up conformations. No specific protein-glycan interactions are found to support the 1Up-to-2Up transition, while more contacts between RBD_B_ and glycan at N165_C_ are observed in the two-Up-like conformations, emphasizing its role as a bottleneck of the transition.”

Reviewer #3 (Recommendations for the authors):This article focuses on the varied conformations that the SARS-CoV-2 spike protein takes and how this relates to the function of this protein and potential for drug targets. Extensive molecular simulations at the atomistic level were performed to provide details of the structural transitions of the spike protein and its protective sugar shield. Overall, this manuscript is a strong study on the spike structural states and transitions using enhanced molecular dynamics. This work clearly lays out details of transitions near the down and up states and how glycans can stabilize or alter conformational states and transitions. This work adds important details to the growing body of literature on the spike protein.

Thank you for giving us positive comments about our work.

The main weakness in this work is the focus on intermediate structures in drug binding. Although this is an interesting idea, the details of the stability of the intermediate states are not described. Stability is key to stable for drug binding to deactivate the spike protein. It is unclear without more detail if these intermediate states are likely to be stable for drug binding and deactivation.

Most of the intermediate states that we detected are considered as stable structures due to the characteristic hydrogen bonds and contacts. As we showed in Figures 4 and 5, I2a, I3a, 1U_L_, 2Ua_L_, and 2Ub_L_, include high probabilities of H-bond or contact pairs, which suggest the important roles on stabilizing the intermediate structures.

The main weakness in this work is the focus on intermediate structures in drug binding. Although this is an interesting idea, the details of the stability of the intermediate states are not described. Stability is key to stable for drug binding to deactivate the spike protein. It is unclear without more detail if these intermediate states are likely to be stable for drug binding and deactivation.Author Recommendations (details):1. Lack of details for intermediate states: I don't see analysis showing intermediate stability of I2a and I3a. How stable are these and where are they on the reduced free energy surface in Figure 1? Short-lived intermediates are not good targets for drugs. Figure 4 provides some details in the structure but no info on the free energy surface near the intermediates is provided and how this relates to stability.

We added a new figure projecting I2a and I3a on PC1-PC2 and Hinge-Hinge free energy maps, Figure 4—figure supplement 11. As shown both intermediates lie in a stable basin as shown by the contour representation or the original FELs. I2a especially seems to be a stable structure along the transition pathway. On page 14, lines 344-346 in the revised manuscript, the following text was added:

“Both I2a and I3a show high stability on the free-energy landscapes on PC1-PC2 and Hinge-Hinge angles as shown in Figure 4—figure supplement 11. Notably, allosteric sites in intermediate structures were previously used for drug design.^61-62^"

2. Accuracy of intermediate state: How do you know these are accurate intermediate states? Any experimental comparison that would suggest these exist?

As we mentioned, any intermediate states have transient nature compared to the stable cryo-EM structures, such as Down or 1Up forms. Therefore, it is difficult for us to compare all the intermediate structures found in the simulations with experimental results in the atomic detail. What we could do is to compare these structures with available cryo-EM structures of wild type, mutants, and spike proteins bound with other molecules, such as a fragment of ACE2 or antibodies.

For the comparison, one cryo-EM structure of D614G mutant (PDB: 7KRS) with one RBD in an intermediate Up conformation is useful. The structure resembles I2a or I3a with Ca RMSD of 4.9 and 5.1 Å, respectively. A new supplementary figure is added, Figure 4—figure supplement 12.

In addition, comparison between the simulation results and smFRET is in good agreement at least quantitatively. Instead, distance distributions calculated from the available cryo-EM structures are missing the distributions between Down and Up structures (Figure 4—figure supplement 4), which suggests transient natures of intermediates between the Down and Up structures and the difficulty to detect these intermediates experimentally.

On page 12, lines 294-298, the following text was added:

Comparison to cryo-EM and smFRET experiment also indicates the quality of the predicted structures including intermediate states. In fact, comparison of the intermediate structures I2a and I3a with the D614G Spike variant cryo-EM structure (PDB:7KRS)^54^ shows a good agreement, with C RMSD of 4.9 and 5.1 Å, respectively (Figure 4—figure supplement 12).”

3. Is spike a good drug target?: The spike protein is certainly an important target for antibodies and vaccines. However, it may not be the best target toward drugs and a current FDA approved drug focuses on another part of the virus action. Some discussion on why the spike protein is believed to be a good target should be included in the manuscript.

Spike has been considered as a primary target for drug repurposing since the start of the pandemic, due to its role in viral entry. Numerous studies have been performed using Spike as drug target, however, to date, these efforts haven’t successful yet. On contrast, despite the general problem in selectivity, targeting proteases was more successful as demonstrated by the new drug from Pfizer. The only drawback of targeting Spike is the high mutation rate in comparison to other proteins in SARS-Cov-2. Unfortunately, such mutation could potentially alter binding site, epitope and vaccine design, as recently shown with Omicron.

On page 14, lines 340-341 in the revised manuscript, the following sentence was added:

“As a consequence of its central role, Spike protein has been early identified as potential target for drug reproposing in order to block viral entry.

On page 15, lines 355-357, the following sentence was added at the end of the paragraph:

“Despite the promising outcome from shifting S-protein equilibrium, the high rate of mutations in Spike might hinder its use as drug and antibodies target or even vaccine development. Indeed, continues assessment of cryptic and epitope regions in Spike variants is necessary.”

4. Supporting figure format: I find it difficult to read and follow the format for the supporting figures that are tied to a main figure and not in a single pdf document. I would hope that eLife allows a single supporting document with all the figures numbered to easily review like all other journals.

We prepared a review only document that includes all the supplementary figures and tables.

5. Add omicron: With the current state of SARS-CoV-2, the list of mutants/variants in the intro should now include omicron.

On page 2, lines 42-44 in the revised manuscript, the following sentence was modified:

“At the same time, more infectious mutants such as B.1.617.2 (Δ), B.1.427 / B.1.429 (Epsilon) and more recently Omicron have appeared^5-6^,”

To know the intrinsic flexibility of spike protein is important for better understanding of the activation mechanisms, i.e. structural transitions from the inactive Down to the active Up conformations. To emphasize this, we added discussion at the end of Discussion section (on page 16, lines 388-392 in the revised manuscript):

“To understand the effect of mutations in the conformational changes toward the active Up form, the difference of intrinsic flexibility between wild type and mutant spike protein is one of the most important features to be understood. The current simulation and analysis methods are applicable to new variants for giving atomistic pictures of conformational flexibility and transitions of spike proteins, which are both important for understanding the essential characters of new spike protein variants.”